# A Unified Causal Framework for Auditing Recommender Systems for Ethical Concerns

## Abstract

As recommender systems become widely deployed in different domains, they increasingly influence their users' beliefs and preferences. Auditing recommender systems is crucial as it not only ensures the improvement of recommendation algorithms but also provides ways to assess and address ethical concerns surrounding them. In this work, we view recommender system auditing from a causal lens and provide a general recipe for defining auditing metrics. Under this general causal auditing framework, we categorize existing auditing metrics and identify gaps in them—notably, the lack of metrics for auditing user agency while accounting for the multi-step dynamics of the recommendation process. We leverage our framework and propose two classes of such metrics: future- and past-reachability and stability, that measure the ability of a user to influence their own and other users' recommendations, respectively. We provide both a gradient-based and a black-box approach for computing these metrics, allowing the auditor to compute them under different levels of access to the recommender system. Empirically, we demonstrate the efficacy of methods for computing the proposed metrics and inspect the design of recommender systems through these proposed metrics.

## 1 Introduction

Recommender systems actively shape people's online experiences by determining which information (e.g., social media posts and job postings) is most visible to them. These socio-technical systems have the potential to influence individuals' choices and beliefs as well as public opinion at large which can have tangible ethical impact (Milano et al., 2020a; Burki, 2019; Rafailidis & Nanopoulos, 2016). Monitoring recommender systems for potential harmful effects is a difficult task, requiring careful design of evaluation metrics and auditing frameworks.

Traditional metrics for evaluating recommender systems are often correlational in nature, focusing on the system's ability to recommend content perceived as relevant by the users based on their past behavior. However, these metrics tend to overlook critical ethical concerns, particularly how these systems interact with and shape user beliefs and preferences over time, raising concerns about user agency and the extent to which users control their own online experiences.

Metrics for measuring user agency are inherently *causal* because they involve answering *what-if* questions. For example, the question "Do users have agency over their recommendations?" requires causal reasoning on the behavior of the recommender system under user behavior shift (Dean et al., 2020; Akp, 2022). Instead of pure prediction, we are interested in the effect of interventions. While some causal auditing metrics have been proposed (Dean et al., 2020; Curmei et al., 2021), a principled framework for reasoning about them in recommender system audits is still lacking.

We present a causal graphical model to capture the dynamics of general recommender systems including matrix factorization and neural network-based models (Section 3.1). This causal model allows us to provide a general recipe for defining causal metrics for auditing recommender systems, and categorize existing auditing metrics based on interventions and outcomes of interest (Section 3.2).

Addressing the lack of metrics for measuring user agency, we use the unified causal framework to formalize two classes of metrics of user agency: past- and future-*reachability* (Section 4.1), which extends the one defined in Dean et al. (2020) as well as past- and future-*stability* (Section 4.2) Reachability measures a user's ability to be recommended (or *reach*) a desired item under modifications to

their future or historical ratings. Stability measures how sensitive a user's recommendations are to edits made by adversarial users to their own ratings.

Next, we provide both a gradient-based and a black-box approach for computing these user-agency metrics. When the recommender system is learned through matrix factorization, under mild assumptions, we show that the corresponding optimization objectives for these metrics have a special structure that can be leveraged to obtain the optimum efficiently (Section 5). Using the proposed metrics, we inspect how the stochasticity of recommendations and the nature of the recommender system (sequence-dependent versus not) may influence user agency. We find, on average, an increase in the stability of a user's recommendations but smaller upgrades in reachability as the stochasticity of the system decreases; and that a standard matrix factorization based recommender system is less stable but facilitates better reachability for users as compared to a deep recurrent recommender network (Section 6). We summarize our contributions as follows:

- We provide a general causal framework for defining new causal metrics and categorizing existing metrics for auditing recommender systems in a principled manner (Section 3);
- Using our proposed framework, we develop two classes of metrics for measuring user agency while accounting for the dynamics of the recommendation process: past- and future-reachability and stability (Section 4). We provide effective ways to compute the metrics, allowing the auditor to have different levels of access to the systems (Section 5).
- Empirically, we investigate two common classes of recommender systems in terms of our proposed user agency metrics and found that higher stochasticity in a recommender system will help with stability but harm reachability (Section 6).

## 2    RELATED WORK

Prior works on ethical issues in recommender systems have identified several areas of concerns including fairness, inappropriate content, privacy, autonomy and personal identity, opacity, and wider social effects (Milano et al., 2020b). A variety of metrics have been proposed to address measures of fairness (Patro et al., 2022; Chen et al., 2020), moral appropriateness of content (Tang & Winoto, 2015), stability (Adomavicius & Zhang, 2012), and diversity (Nguyen et al., 2014; Silveira et al., 2019; Parapar & Radlinski, 2021). Most of these metrics rely on observational quantities under a fixed recommendation policy and neglect the effects of users' behaviors, preferences, or the recommender system itself. Studying user agency requires answering causal questions pertaining to how a recommender would respond to interventions made in the user's behavior over time, which cannot be answered by association-based metrics alone. Therefore, the metrics we formalize are interventional and counterfactual. Interventional metrics quantify the effect of hypothetical changes to specific parts of the recommender system. Counterfactual metrics evaluate how the system would behave if interventions were performed in hindsight while taking into account what actually happened.

Recently there have been works taking a step towards defining interventional metrics (e.g., (Dean et al., 2020; Curmei et al., 2021; Chen et al., 2020)) by considering the effect of specific interventions on the recommender system's behavior. Dean et al. (2020); Curmei et al. (2021) take the intervention to be users' own feedback and the outcome of interest to be the recommendation they receive, allowing their proposed metric to answer questions concerning user agency. However, existing causal metrics are often limited in the kind of interventions they consider, with few works tapping into new types of interventions and ethical concerns. In addition, most are one-step metrics, ignoring the recommender-user interaction dynamics over time. Separately, another line of work integrates causal approaches into recommender systems, often proposing new causality-based models focusing on estimating the direct effect of recommendations on user engagement (Schnabel et al., 2016; Sharma et al., 2015; Sato et al., 2020). These metrics fall under the interventional layer of Pearl's causal hierarchy, and choose the intervention to be the recommendations themselves, and the outcome of interest in this case is commonly the user feedback (Chen et al., 2020).

## 3    A CAUSAL PERSPECTIVE ON AUDITING RECOMMENDER SYSTEMS

We first provide a general recommender system setup (Section 3.1), its corresponding causal graph (Figure 1), and an example illustrating the general setup. We then contextualize existing metrics for

auditing recommender systems using this causal framework and provide a general recipe for one to develop new auditing metrics (Section 3.2).

## 3.1 RECOMMENDER SYSTEM SETUP

We consider the setting where there are $n$ users and a set of items $\mathcal{V}$ to recommend. Each user $i \in [n]$ has an unobserved vector $\mathbf{o}_i^\star \in \mathbb{R}^{|\mathcal{V}|}$ indicating the user's initial (true) preference (i.e., ratings) for all items $\mathcal{V}$. At each time step $t \in [T]$, a recommender system presents a recommendation (set) $\mathbf{A}_{i,t} \subseteq \mathcal{V}$ to user $i$. In turn, the user provides their feedback/ratings $\mathbf{O}_{i,t} \in (\mathbb{R} \cup \{\text{unrated}\})^{|\mathcal{V}|}$ for the recommendation where $\mathbf{O}_{i,t}[k]$ is the $k$-th entry of user's rating vector, $\mathbf{O}_{i,t}[k] = \text{unrated}$ for $k \notin \mathbf{A}_{i,t}$, and $\mathbf{O}_{i,t}[k] \in (\mathbb{R} \cup \{\text{unrated}\})$ indicates user's choice of rating or not rating for item $k \in \mathbf{A}_{i,t}$. The user-recommender interaction history is denoted by $\mathbf{H}_{i,t} = (\mathbf{A}_{i,1}, \mathbf{O}_{i,1}, \cdots, \mathbf{A}_{i,t-1}, \mathbf{O}_{i,t-1})$ and we use $\mathbf{D}_t = \{\mathbf{H}_{i,t}\}_{i \in [n]}$ to denote the interaction history for all users, or in other words, the training dataset for the recommender at time $t$. It is worth noting that both $\mathbf{A}_{i,t}$ and $\mathbf{O}_{i,t}$ are random variables depending on previous user-recommender interactions. More specifically, the recommendation $\mathbf{A}_{i,t}$ depends on the training set $\mathbf{D}_t$; the user rating $\mathbf{O}_{i,t}$ is a function of the user's true initial preference $\mathbf{o}_i^\star$, their interaction history $\mathbf{H}_{i,t}$, and possibly other users' feedback in $\mathbf{D}_{-i,t}$. Finally, we let $\mathbf{D}_{-i,t}$ denote $\mathbf{D}_t \setminus \mathbf{H}_{i,t}$ and use $\mathbf{O}_{-i}$ to refer to user ratings for users $[n] \setminus \{i\}$. When clear from the context, we omit the subscript $i$ or $t$ for discussing the corresponding quantity for all users or time steps. For any variable or function $y$, we use $y_{t_1:t_2}$ to refer to $y_{t_1}, \cdots, y_{t_2}$.

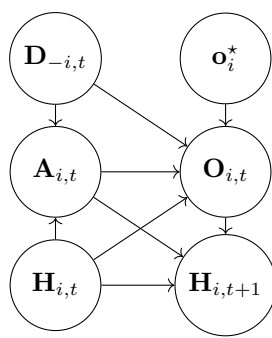

Figure 1: Causal graph representing a general recommender system at time $t$, specifically pertaining to a user $i$'s interactions with the system. Here, $D_{-i,t}$ denotes the interaction history of all users besides $i$ upto $t$, $A_{i,t}$ denotes the set of recommendations $i$ receives at $t$, $H_{i,t}$ denotes $i$'s interaction history upto $t$, $O_{i,t}$ denotes $i$'s feedback at this time and $o_i^*$ denotes $i$'s true preference.

We use the Structural Causal Model (SCM) framework (Pearl, 2009) to capture the causal relationships among these variables, with further details provided in Appendix A. The causal graph for the recommender system is given in Fig. 1. We provide a concrete example:

**Example 1.** *In the basic form of a score-based recommender system, the initial preference vector $\mathbf{o}_i^\star$ is the user's true ratings for item set $\mathcal{V}$, the recommendation $\mathbf{A}_{i,t}$ is a singleton[*], and the user rating is given by $\mathbf{O}_{i,t} = \mathbf{o}_i^\star[\mathbf{A}_{i,t}]$. To determine the recommendation $\mathbf{A}_{i,t}$, a recommendation algorithm first predicts a score per user-item pair $(i, j)$, indicating the predicted user preference on item $j \in \mathcal{V}$ for user $i \in [n]$. To learn the scoring function, one rely on the dataset $\mathbf{D}_t$ and may use approaches like matrix factorization (more details in Section 5). Finally, given the score, a recommendation algorithm may recommend differently, e.g., deterministically according to the highest score on unseen items for the user, or stochastically among the top scoring items.*

From an auditor's perspective, one may be interested in defining metrics to measure how the ratings among users correlate with each other (associational metrics), how the recommendation algorithm may cause the popularity of an item (interventional metrics), or how much change a user can make over their recommendations had they behaved differently under the same historical recommender (counterfactual metrics). Hereafter, we provide a discussion of associational, interventional, and counterfactual metrics for auditing a recommender system, categorizing existing metrics using this causal framework, and illustrating how one can use this unifying perspective to define new metrics.

## 3.2 CAUSAL PERSPECTIVE ON AUDITING RECOMMENDER SYSTEMS

Existing auditing metrics often focus on the *association* between variables concerning different qualities of recommendations under the current system. Metrics like item diversity, average ratings, or popularity, are of this kind since they are defined over the observational distribution $\mathbb{P}(\{\mathbf{A}_{i,t}, \mathbf{O}_{i,t}\})$ (e.g., (Abdollahpouri et al., 2019; Rastegarpanah et al., 2019)).

---

[*]With some abuse of notation, we say $\mathbf{A}_{i,t} \in \mathcal{V}$ in this case.

When defining *interventional* and *counterfactual* metrics, one needs to be explicit about the intervention to apply, the outcome one cares about, and the distributions involved in obtaining the metric. More specifically, to define a metric of causal nature, one needs to specify the following three quantities of interests: *(i)* the intervention; *(ii)* the (random) outcome of interest; and *(iii)* a metric (or in other words, a functional) that maps the random outcome of interest to a real value. This process allows us to categorize existing causal metrics used for auditing recommender systems based on the nature of the intervention and outcome of interest:

- Recommendation $\mathbf{A}_i$ as intervention: In this case, the auditor interrogates the effect of changing the recommendation algorithm. Often, the outcome of interest is user ratings $\mathbf{O}_i$, and the corresponding interventional distribution is $\mathbb{P}(\mathbf{O}_i|\text{do}(\mathbf{A}))$ (e.g., exposure bias (Chen et al., 2023)). This quantity is often at the core of a recommendation algorithm and is studied the most, as recommender systems use predicted user preferences to make recommendations, through optimizing the recommendations against user interests, e.g., $\max_{\mathbf{a}_i} \mathbb{P}(\mathbf{O}_i|\text{do}(\mathbf{A} = \mathbf{a}_i))^{\dagger}$.
- Other user's data $\mathbf{D}_{-i}$ as intervention: Here, the auditor may want to investigate under the current recommender system, what would happen to a particular user (or a particular group of users) if other users have changed their behavior. For example, in measuring conformity bias (i.e., how other users' ratings may influence the user's reaction to the same item) (Chen et al., 2023), the auditor inspects whether the probability of the user's own rating change as one intervenes on other user's historical ratings/data. The corresponding interventional distribution is $\mathbb{P}(\mathbf{O}_i|\text{do}(\mathbf{D}_{-i}))$.
- User's own reaction $\mathbf{O}_i$ (or $\mathbf{H}_i$) as intervention: The auditor may be interested in inspecting the amount of change a user's own reaction may cause in their recommendations. In this case, The outcome of interest, as we will discuss in Section 4, can be the user's recommendation (Dean et al., 2020), i.e., the distribution of interest is $\mathbb{P}(\mathbf{A}_i|\text{do}(\mathbf{O}_i))$.

With this perspective and the proposed three-step recipe for defining new metrics, an auditor can identify gaps in existing metrics and define new ones. As an example, in Section 4, we define a suite of user-centric causal metrics using this framework. Finally, there is often less discussion on counterfactual metrics in auditing recommender systems, since it involves inspecting quantities that are hard (if not impossible) to obtain in practice. We provide more discussion on this in Section 4. We also note that the aforementioned metrics are only example metrics that one can categorize using this framework. There are other metrics that fit in this framework (e.g., selection bias and position bias (Chen et al., 2023)) that we do not go into details.

## 4 USER-CENTRIC CAUSAL METRICS FOR AUDITING USER AGENCY

While much attention has been given to auditing various ethical concerns of recommender systems, there has been comparatively little work conducted on measuring user agency. This gap is particularly notable given the rich line of qualitative work emphasizing the importance of user agency (Milano et al., 2020a). User agency is a user's power over their own recommendations compared to recommendations being driven by external forces like other users' behaviors or algorithmic profiling (de Vries, 2010). It can be compromised in various ways, several of which we target with the metrics we propose. First, recommender systems can enforce filter bubbles that restrict users from diverse content feeds and amplify biases (Milano et al., 2020b). Second, recommendation algorithms can be vulnerable to strategic behavior and adversarial attacks that alter recommendations for unrelated users (Milano et al., 2020b). For example, consider content duplication attacks on e-commerce platforms (Fröbe et al., 2020). In this setting, providers game the recommendation system by duplicating item listings with little to no changes to maximize the probability of recommendation. Maintaining user agency over their own recommendations in this scenario requires recommendation stability. The user-centric causal metrics we define in this section are designed with the intent of quantifying user agency from these two separate viewpoints: reachability measures the extent to which a user can escape filter bubbles (Section 4.1), while stability assesses the influence that other strategic users can have on this user's recommendations (Section 4.2).

When defining these metrics, we take the interventions as user's feedback $\mathbf{O}_i$ or other user's feedback $\mathbf{O}_{-i}$, and the outcome of interests as their recommendations $\mathbf{A}_i$. We call these metrics user-centric since the interventions are user feedback (actions that users can take), and the outcome of interest is

---

$^{\dagger}$On a separate note, an active line of work has centered around using observational data to estimate this quantity, e.g., Schnabel et al. (2016).

the user's own recommendations (that are consumed by the user). In Section 4.1, we extend existing metrics for auditors to measure the autonomy an user can have regarding their own recommendations; In Section 4.2, we develop new metrics for auditors to measure the influence a user's choice can have on other users' recommendations. Throughout, we provide both interventional and counterfactual metrics and discuss how these metrics connect to the dynamics of the recommendation process.

## 4.1 PAST- AND FUTURE-REACHABILITY

A recent line of work develops metrics for inspecting user agency, with the most relevant work introducing the concept of (maximum) stochastic recheability (Curmei et al., 2021; Dean et al., 2020). Conceptually, stochastic reachability measures the maximum change in recommendation probability on item $j$ for user $i$ upon modifying their ratings for a predefined set of items $\mathbf{a}$, e.g., historical recommendations or unseen items for the user. Under our framework, the original maximum stochastic recheability for a user-item pair $(i, j)$ at time $t$ can be rewritten as:

$$\max_{f \in \mathcal{F}} \mathbb{E}_{\mathbf{A}_{i,t}} \left[ \mathbb{P}(\mathbf{A}_{i,t+1} = j | \mathrm{do}(\mathbf{O}_{i,t} = f(\mathbf{A}_{i,t}))) \right],$$

where $\mathcal{F}$ is the set of all measurable functions that map a recommendation set $\mathbf{a}$ to a user rating vector $\mathbf{o}$, and $\mathbf{a}$ is a fixed set of (recommendation) items that can be randomly generated from the unseen or historically recommended set. By rewriting the original stochastic reachability using our framework, we identify that the original metric allows the auditor to inspect user agency for a single time step and the intervention is specific to user ratings for a particular set of predefined recommendations $\mathbf{a}$ (that are not determined by the dynamics of the recommendation process). In does not allow an auditor to inspect user agency under the natural evolution of a recommender system as users apply interventions over time. To this end, we introduce the following (maximum) future-$k$ reacheability.

**Definition 4.1** (future-$k$ reachability). *For any user $i \in [n]$ and item $j \in \mathcal{V}$, at time $t \in \mathbb{N}_+$, their maximum future-$k$ reachability for $k \in \mathbb{N}_+$ is given by*

$$\max_{f_{1:k} \in \mathcal{F}^k} \mathbb{E}_{\mathbf{A}_{i,t:t+k-1}} \left[ \mathbb{P}(\mathbf{A}_{i,t+k} = j | do(\mathbf{O}_{i,t} = f_1(\mathbf{A}_{i,t})), \ldots, do(\mathbf{O}_{i,t+k-1} = f_k(\mathbf{A}_{i,t+k-1}))) \right], \tag{1}$$

*where the expectation is over the recommendation trajectory $\mathbf{A}_{i,t:t+k-1}$ and $\mathcal{F}^k = \mathcal{F} \times \cdots \times \mathcal{F}$.*

In this case, the intervention $f_{t'}(\mathbf{A}_{i,t+t'-1})$ would affect the distribution of $\mathbf{A}_{i,t+t'}$, the recommendations in the next time step. This metric formalizes a notion of reachability where the user can arbitrarily change the rating of the recommended item, but the item trajectory (the sequence of items they choose to rate) follows the (future) dynamics of the recommender system itself.

In parallel, we define another metric that quantifies reachability under edits to the history over the past $k$ time steps, thus allowing the auditor to inspect user agency retrospectively. Consider a recommender system at time $t$, that we call the *factual* recommender system. We denote the *factual* trajectory for a user $i$ over the past $k$ time steps by $\mathbf{h}^{\star}_{i,t-k:} = (\mathbf{a}^{\star}_{i,t-k}, \mathbf{o}^{\star}_{i,t-k}, \cdots, \mathbf{a}^{\star}_{i,t-1}, \mathbf{o}^{\star}_{i,t-1})$. Now, consider a *counterfactual* recommender system that is retrained at time $t$ by modifying the ratings in $\mathbf{h}^{\star}_{i,t-k:}$ to a counterfactual history $\mathbf{h}_{i,t-k:}$, while keeping everything else the same as the *factual* recommender system.

**Definition 4.2** (Past-$k$ reachability). *For any user $i \in [n]$ and item $j \in \mathcal{V}$, at time $t \in \mathbb{N}_+$, their maximum past-$k$ reachability for $k \in \mathbb{N}_+$ is given by*

$$\max_{f_{1:k} \in \mathcal{F}^k} \mathbb{E}_{\mathbf{h}^{\star}_{i,t-k:}} \left[ \mathbb{P}(\mathbf{A}_{i,t} = j | do(\mathbf{H}_{i,t-k:} = \mathbf{h}_{i,t-k:})) \right], \tag{2}$$

*where $\mathbf{h}_{i,t-k:} = (\mathbf{a}^{\star}_{i,t-k}, f_1(\mathbf{a}^{\star}_{i,t-k}), \cdots, \mathbf{a}^{\star}_{i,t-1}, f_k(\mathbf{a}^{\star}_{i,t-1}))$, $\mathbf{A}_{i,t}$ denotes the counterfactual recommendation under the edited history $\mathbf{h}_{i,t-k:}$. The expectation is taken over the 'factual' history $\mathbf{h}^{\star}_{i,t-k:}$ and the edited history $\mathbf{h}_{i,t-k:}$ depends on the factual history $\mathbf{h}^{\star}_{i,t-k:}$*

This metric is a counterfactual quantity (Pearl, 2009, Ch. 7) as it involves random variables from both the factual and counterfactual recommender systems. Informally, in past-$k$ reachability, we maximize the counterfactual recommendation probability where the item trajectory is fixed to the factual trajectory, but the ratings can be arbitrarily edited. By contrast, in the future-$k$ metric, the trajectory follows the recommender dynamics (instead of being set to the factual trajectory).

## 4.2 PAST- AND FUTURE-(IN)STABILITY

We introduce a new class of metrics that allow the auditor to measure the influence other users have on a particular user's recommendations. In addition to their own feedback and the recommendation algorithm itself, a user's autonomy is also influenced by other users' feedback to the recommender systems. The class of (in)stability metrics we define below allows auditors to inspect user agency through this angle.

**Definition 4.3** (Future $k$-(In)stability). *Given user $i_1, i_2 \in [n]$ such that $i_1$ is the user of interest and $i_2$ is the user who can update their feedback, at time $t \in \mathbb{N}_+$, user $i_1$'s future-k maximum instability with respect to user $i_2$ for $k \in \mathbb{N}_+$ is given by*

$$
\max_{f_{1:k} \in \mathcal{F}^k} \mathbb{E}_{\mathbf{A}_{i_2,t:t+k-1}} \Bigg[ d(\mathbb{P}(\mathbf{A}_{i_1,t+k}|
$$

$$
do(\mathbf{O}_{i_2,t} = f_1(\mathbf{A}_{i_2,t})), \dots, do(\mathbf{O}_{i_2,t_0+k-1} = f_k(\mathbf{A}_{i_2,t+k-1}))), \mathbb{P}(\mathbf{A}_{i_1,t+k})) \Bigg], \quad (3)
$$

*where $d : \Delta \times \Delta \to \mathbb{R}_{\geq 0}$ measures the distance between two probability distributions.*

In this case, an auditor seeks to intervene on the ratings given by user $i_2$ and observe how these interventions affect the recommendations received by user $i_1$, following the dynamics of the recommendation process. The metric is defined as the deviance between the recommendation distribution before and after the intervention. This allows the auditor to measure how unstable (or in another way, how manipulatable) the recommendation for user $i_1$ can be with respect to user $i_2$'s feedback.

Similar to the past-$k$ reachability metric (see Defn. 4.2), we now define a counterfactual past-$k$ stability metric which quantifies the stability of user $i_1$ under edits to the history of user $i_2$. Informally, we seek to maximally change the recommendation probability of item $j$ for user $i_1$ by editing the ratings of user $i_2$, while keeping their item trajectory to the factual one.

**Definition 4.4** (Past $k$-(In)stability). *Let $\mathbf{h}^\star_{i_2,t-k:} = (\mathbf{a}^\star_{i_2,t-k}, \mathbf{o}^\star_{i_2,t-k}, \cdots, \mathbf{a}^\star_{i_2,t-1}, \mathbf{o}^\star_{i_2,t-1})$ denote the* factual *recommendation history for user $i_2$. We define user $i_1$'s past-k maximum instability with respect to user $i_2$ for $k \in \mathbb{N}_+$ as*

$$
\max_{f_{1:k} \in \mathcal{F}^k} \mathbb{E}_{\mathbf{h}^\star_{i_2,t-k:}} \left[ d\left(\mathbb{P}(\mathbf{A}_{i_1,t} = j | do(\mathbf{H}_{i_2,t-k:} = \mathbf{h}_{i_2,t-k:})), \mathbb{P}(\mathbf{A}_{i_1,t} = j)\right) \right], \quad (4)
$$

*where $\mathbf{h}_{i_2,t-k:} = (\mathbf{a}^\star_{i_2,t-k}, f_1(\mathbf{a}^\star_{i_2,t-k}), \cdots, \mathbf{a}^\star_{i_2,t-1}, f_k(\mathbf{a}^\star_{i_2,t-1}))$ is the edited counterfactual history (which depends on the factual history $\mathbf{h}^\star_{i_2,t-k:}$), and $\mathbf{A}_{i_1,t}$ is the counterfactual recommendation.*

While both interventional and counterfactual metrics measures different aspects of user agency, there is a distinction in their scope:

- Past-/counterfactual metrics focus on how user behavior (e.g., the items a user chose to rate) contributes to the recommendations they receive in the present. For example, consider a social media user who primarily receives recommendations for cat videos in the present. Counterfactual metrics help us understand whether the narrowness in the recommendations can be attributed to the recommendation system, or the user's behavior in the past, which imply vastly different conclusions in terms of user agency. If engaging with cat videos unfavorably in the past would have led to more diverse recommendations in the present, the observed narrow recommendations do not imply a violation of user agency.
- Future-/interventional metrics focus on the user's radius of influence on recommendations in the future. Assume the user who likes cat videos adjusts their preferences and now shows interest in dog videos. Interventional metrics help us understand the ability of changed user behavior to steer upcoming recommendations. If alterations in behavior (engaging with more dog videos) do not lead to diversification of recommendations away from only cat content, we can conclude that there is a violation of user agency for their own recommendations.

Though both reachability and stability are measures of user agency, the two metrics focus on different aspects of it: stability focuses on the influences of behavior changes of other users, while reachability focuses on influences of the individual at hand. Empirically, in our experiments with embedding-based recommender systems, we observe a trade-off between these two metrics (Section 6). Further investigation into this trade-off is an interesting direction for future work.

In the following, we provide an operationalized procedure for computing these user-centric metrics; the exact implementation details can be found in Section 6 and Appendix D.

## 5 OPERATIONALIZED PROCEDURE FOR COMPUTING USER AGENCY METRICS

When computing the user-centric causal metrics, we need to consider two aspects: *(i)* whether the auditor can intervene on the recommender system, i.e., if they have access to the interventional distribution; and *(ii)* whether they can obtain gradients of the objective functions in equation 1-equation 4. By obtaining the gradients, the metrics can be computed through methods like gradient descent. Similar to Dean et al. (2020), we consider cases where the auditor can apply intervention on the recommender system, thus addressing concerns around *(i)*. We provide methods that allow auditors to have different levels of access to the internals of the system (e.g., they may not directly have access to the required gradients). Before delving into our methods, we first provide a high-level discussion on the required gradients. Consider the case where the auditor aims to compute equation 1 when $k = 1$, we need

$$\nabla_{f_1} \mathbb{E}_{\mathbf{A}_{i,t}} \left[ \mathbb{P}(\mathbf{A}_{i,t+1} = j | \mathrm{do}(\mathbf{O}_{i,t} = f_1(\mathbf{A}_{i,t}))) \right]$$

$$= \sum_{\mathbf{a}_{i,t}} \mathbb{P}(\mathbf{A}_{i,t} = \mathbf{a}_{i,t}) \nabla_{f_1} \mathbb{P}(\mathbf{A}_{i,t+1} = j | \mathrm{do}(\mathbf{O}_{i,t} = f_1(\mathbf{a}_{i,t}))),$$

where $\nabla_{f_1}$ refers to gradient with respect to the parameters of $f_1$ and the equality holds because of Fubini's theorem. In this case, the auditor needs to effectively compute $\nabla_{f_1} \mathbb{P}(\mathbf{A}_{i,t+1} = j | \mathrm{do}(\mathbf{O}_{i,t} = f_1(\mathbf{a}_{i,t})))$ for all $\mathbf{a}_{i,t} \subset \mathcal{V}$. In the following, we first discuss cases where the auditor has access to the actual gradients. We show that under mild conditions, for matrix factorization based recommender systems, the objective in equation 2 is concave and equation 4 is quasi-convex and has its optimum at extreme points (Section 5.1). For cases where the auditor only has access to zeroth-order information of the system, we provide an effective approach for approximating the required gradient (Section 5.2). We show simplified form of the gradients we require access to for future-$k$ reachability/stability in Appendix G, similar to the one step gradient above.

### 5.1 WHITE-BOX ACCESS

In the most desirable setting, the auditor has what we term as white-box access to the recommender system. In this case, the auditor has access to the gradient of the interventional distribution of interest. We consider two types of score-based systems where the recommendation probability is given by

$$\mathbb{P}(\mathbf{A}_{i,t+1} = j | \mathrm{do}(\mathbf{O}_{i,t} = f_1(\mathbf{a}_{i,t}))) \propto \exp(\beta \mathbf{p}_i^\top \mathbf{q}_j), \tag{5}$$

where $\beta > 0$ controls the stochasticity of the system, the user embedding $\mathbf{p}_i$ and item embedding $\mathbf{q}_j$ depend on the intervention $\mathbf{O}_{i,t}$ (and $\mathbf{O}_{-i,t}$). The two types of systems learn $\mathbf{p}_i$ and $\mathbf{q}_j$ differently—one uses matrix factorization and the other uses an LSTM-based approach. For more details, we refer the readers to Appendix B.1.

To obtain the gradient $\nabla_{f_1} \exp(\beta \mathbf{p}_i^\top \mathbf{q}_j)$, we need to know how the systems updates their user and item embeddings when user ratings change. For matrix factorization models, when computing reachability (i.e., when users change their own ratings), we assume item embedding $\{\mathbf{q}_j\}_{j \in \mathcal{V}}$ are fixed and user embedding $\mathbf{p}_i$ is updated according to the matrix factorization objective. Effectively, we are assuming that the user's own ratings only affects their own embedding. The closed form of the update can be found in Appendix B.2. Under this assumption, we have the below result.

**Proposition 5.1.** *In matrix factorization, the past-$k$ reachability objective (equation 2) is concave in the parameters of $f_{1:k}$ if item embeddings $\{\mathbf{q}_j\}_{j \in \mathcal{V}}$ are fixed and $\mathbf{p}_i$ is updated according to the matrix factorization objective when the ratings $\mathbf{O}_i$ change.*

For computing stability (i.e., when other users change their ratings), we assume the user embedding $\{\mathbf{p}_i\}_{i \in [n]}$ are fixed and item embedding $\mathbf{q}_j$ is updated according to the matrix factorization objective. Effectively, we are assuming that other users affect user $i$'s recommendation through item embeddings. The closed form of the update can be found in Appendix B.2. Under this assumption, we have that past-$k$ stability is obtained at boundary points.

**Proposition 5.2.** *In matrix factorization, the past-$k$ stability objective (equation 4) is quasi-convex in the parameters of $f_{1:k}$ and achieves its optimal value at a boundary point of the domain, if user embeddings $\{\mathbf{p}_i\}_{i \in [n]}$ are fixed and item embeddings $\{\mathbf{q}_j\}_{j \in \mathcal{V}}$ are updated according to the matrix factorization objective when user $i_2$'s ratings $\mathbf{O}_{i_2}$ are changed, $d$ is the $L_2$ distance.*

We note that the definitions of the proposed metrics don't require any specific assumptions on how the recommender is trained and we only consider one fixed embedding here for ease of operationalization, borrowing this approach from (Yao et al., 2021; Curmei et al., 2021; Dean et al., 2020). In our experiment, we also consider user embedding $\mathbf{p}_i$ and item embedding $\mathbf{q}_j$ to be the output of an LSTM that takes in user and item history, respectively. In this case, if the auditor has the corresponding LSTM, they may obtain gradient of $\mathbf{p}_i$ and $\mathbf{q}_j$ with respect to user history (Section 6).

## 5.2 BLACK-BOX ACCESS

In certain cases, the auditor can only obtain the output of a recommender through querying the system with different inputs, but no direct access to the recommendation mechanism itself (thus no access to the required gradients). In this case, we leverage the traditional toolkit to estimate the gradient using finite difference methods (e.g., (Malladi et al., 2023)):

$$\nabla_\theta \mathbb{P}(\mathbf{A}_{i,t+1} = j | \mathrm{do}(\mathbf{O}_{i,t} = f_\theta(\mathbf{a}_{i,t})))$$
$$\approx \frac{1}{2\epsilon} |\mathbb{P}(\mathbf{A}_{i,t+1} = j | \mathrm{do}(\mathbf{O}_{i,t} = f_{\theta+\epsilon\mathbf{z}}(\mathbf{a}_{i,t}))) - \mathbb{P}(\mathbf{A}_{i,t+1} = j | \mathrm{do}(\mathbf{O}_{i,t} = f_{\theta-\epsilon\mathbf{z}}(\mathbf{a}_{i,t})))|,$$

where $\theta \in \mathbb{R}^d$, $\epsilon > 0$ is a perturbation scale, and $\mathbf{z} \sim \mathcal{N}(0, \mathbf{I}_d)$. For more details on using zeroth-order information for neural network optimization, we refer the readers to (Malladi et al., 2023).

## 6 EXPERIMENTS

We conduct experiments to show how our proposed metrics of reachability and stability vary for different recommender systems, across different time horizons and as the stochasticity of the recommender system changes. Since these metrics are meant to be proxies of user agency, we show how our the aforementioned experimental parameters affect user agency, by observing how these measurements change as we change these parameters.

### 6.1 EXPERIMENTAL SETUP

**Dataset.** Our experiments are based on the publicly available MovieLens-1M Dataset (Harper & Konstan, 2015) which contains 1,000,209 total ratings given by 6,040 users on 3,706 movies (Table I includes dataset summary statistics). The ratings are integer values between 1 and 5, both inclusive.

**Recommender Systems.** We perform experiments with a Matrix Factorization (MF) recommender system (Koren et al., 2009) from Surprise(Hug, 2020) and a Recurrent Recommender Network (Wu et al., 2017) (additional details about these recommender systems and implementation details on both reachability and stability are included in Appendix D). For future-facing metrics, we use a *deterministic* item selection policy that always recommends the top-1 item. For past-facing metrics, we use the *stochastic* softmax policy in equation 5.

**Experimental Parameters.** There are two main parameters for the metrics:

- Time Horizon $t$: We compare $t = 1$ and $t = 5$ to demonstrate how longer time horizons increase both user 'reach' to items and adversarial manipulation of recommendations. While we tested $t = 1, 2, 3, 4, 5$, we present $t = 1$ vs $t = 5$ to highlight the contrast between short and long-term user agency effects.
- Stochasticity ($\beta$):We compare $\beta = 0.8$ and $\beta = 5$ to illustrate how higher $\beta$ values increase both user 'reach' to items and potential for adversarial manipulation. While we tested multiple $\beta$ values, we present these two to highlight the contrast between short and long-term user agency effects.

Additional plots with different experimental parameters can be found in the Appendix F.

### 6.2 COMPARISON AMONG METRICS AND ALGORITHMS

**Algorithm comparisons.** We demonstrate that zeroth order optimization techniques with black-box access as described in Section 5 can serve as a viable optimization strategy. We use the ratio between the final probability of an item being recommended to a user after intervention and the initial probability of the item being recommended to the user before any intervention to measure the

gain in reachability. This has been referred to as "Lift" in (Curmei et al., 2021), while the initial probability of the item being recommended to the user before any intervention is termed "baseline reachability." We compare the mean values of Lift obtained for both past-5 and future-5 reachability and stability using gradient descent with those obtained using black box access (see Figure 2a for these comparisons). For reachability, gradient descent (GD) outperforms black box access, but black box access still obtains a solution with Lift $> 1$. For stability, black box access converges to the same solution as GD. For stability, we also plot the value obtained by exhaustively searching the extreme points of the rating domain, represented by "Oracle" in 2b.

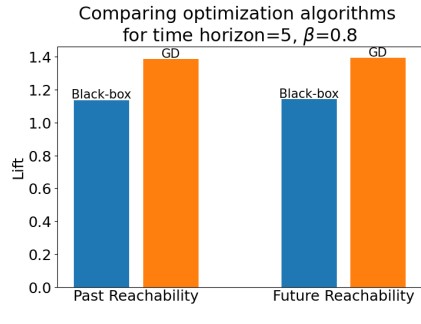 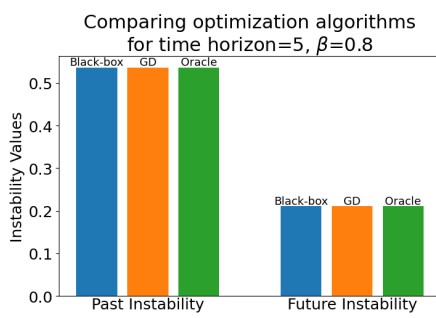

(a) Mean value of reachability lift  (b) Mean values of instability

Figure 2: Computing reachability (2a) and stability (2b) with black-box or gradient (GD) access.

**Metric comparisons.** Allowing both past and future edits for longer time horizons lead to higher gains in reachability compared to shorter time horizons as the user has more freedom to change their ratings. Meanwhile, allowing edits over a longer time horizon decreases the stability of a user's recommendations on average. We observe that most user-adversary pairs have instability values close to 0 or 1, indicating a duality where a user's recommended list is either heavily affected by the actions of an adversary or is minimally affected by them. Figure 3 shows these results.

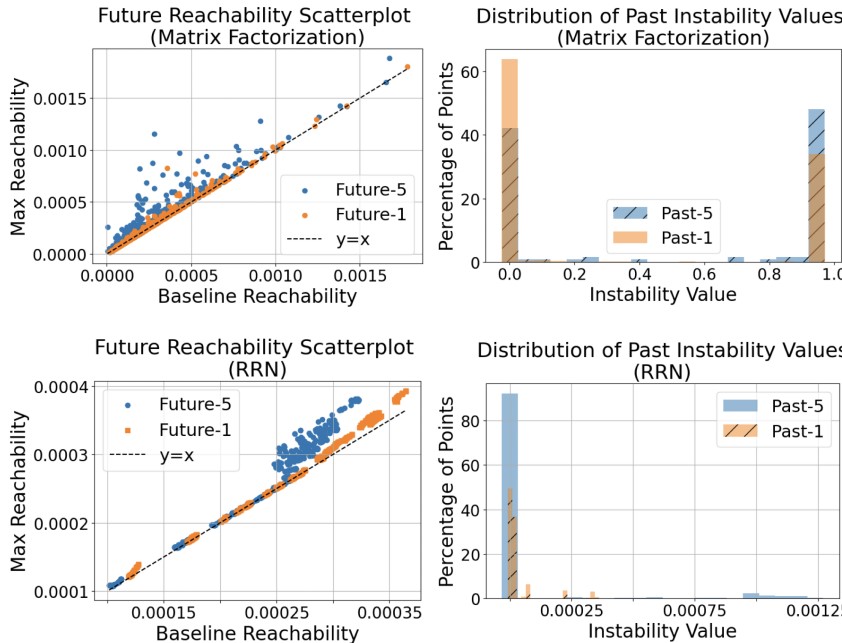

Figure 3: Comparisons between future reachability and past instability for two different values of time horizon for both MF and RRN. Both reachability and instability increase with longer horizons.

### 6.3 INVESTIGATING THE DESIGN OF RECOMMENDER SYSTEMS

**Stochasticity.** A higher $\beta$ leads to less stochastic recommendations. The plot in Figure 4 shows the past-5 reachability of multiple user-item pairs for $\beta = 0.8$ and $\beta = 5$. We observe higher values of

Lift (gain in reachability) for lower stochasticity. Moreover, these values are higher for items with high baseline reachability values in both cases, as the points move away from the $y = x$ line for higher baseline reachability. For $\beta = 5$, we observe multiple Lift values of the order of $10^3$ but for $\beta = 0.8$, we only see lift values slightly greater than 1 for the most part, or of the order of 10 at best.

As we increase $\beta$, or decrease the stochasticity of the system, user recommendations tend to become more stable, as shown in Figure 5a where we plot mean values of instability for multiple user-adversary pairs and varying $\beta$ from 0.2 to 6 in discrete steps. The intuition for this is that a lower stochasticity implies that the user is pushed more towards deterministic recommendations, and in the most extreme case, their recommendation list is perfectly stable, with the entire probability distribution of recommendation being concentrated on one item. We also plot histograms for instability values (Figure 5b) at $\beta = 0.8$ and $\beta = 5$ and see that a greater percentage of items for $\beta = 5$ have their instability values concentrated around 0 than do for $\beta = 0.8$, which matches the prior observation.

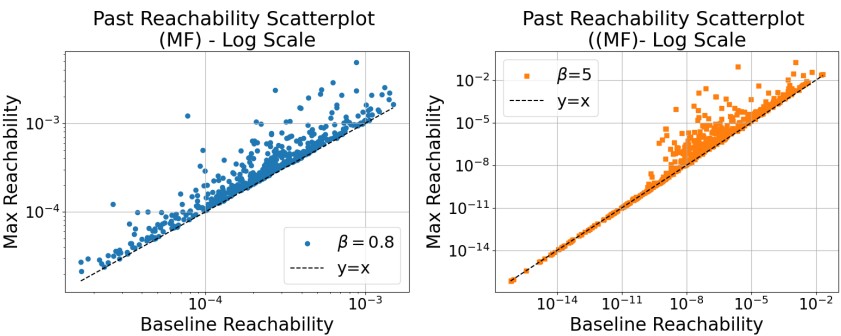

Figure 4: Scatterplot of Past-5-Reachability for a MF based recommender as $\beta$ varies.

**Nature of Recommender Systems** We observe that the MF-based recommender system has higher reachability than the RNN one when averaged over a large number of user-item pairs. On the other hand, the RNN consistently has higher stability than the MF-based system when averaged over a large number of (user-user) pairs. The differences in the two rows in Figure 3 illustrate this, as the reachability scale for the RNN is lower than that for the MF recommender and there is no concentration of instability values around 1 like there is for the MF recommender. We infer that the MF-based system facilitates more reachability but is less stable than its RNN counterpart.

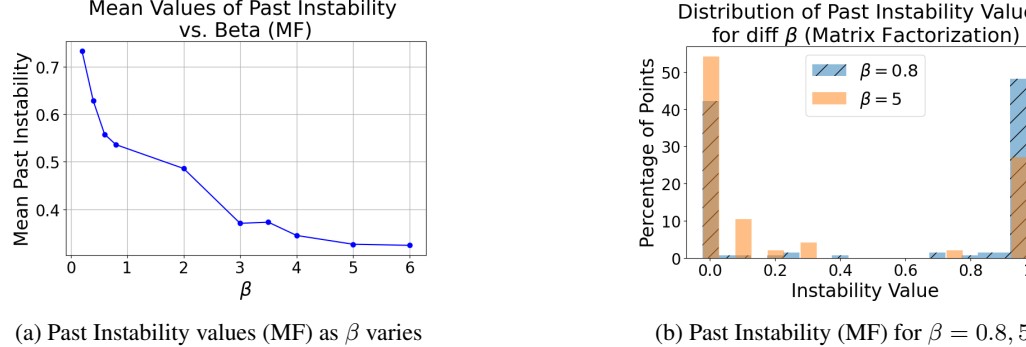

(a) Past Instability values (MF) as $\beta$ varies      (b) Past Instability (MF) for $\beta = 0.8, 5$.

Figure 5: Effect of varying stochasticity on past-stability for a MF recommender system.

**Conclusion.** This paper presents a causal framework to formalize interventional and counterfactual metrics to audit recommender systems in a principled manner. Addressing the problem of user agency, we present the metrics of *reachability* and *stability* to quantify a user's agency over their recommendations under edits to their own or a different user's history. In our experiments, we find that with more stochasticity, users' agency towards their own recommendation increases as indicated by the increase of reachability. However, higher stochasticity also allows other users' to potentially have more control over the systems, indicated by lower stability of the system. Comparing recommendation algorithms, we observed that MF has higher reachability than the RNN, suggesting that more complex models may allow less user agency in some aspects, which could be attributed to the fact that small changes to a dataset have low influence on large-scale deep learning systems.

**Ethics Statement:** Our paper develops a causal framework for auditing recommender systems for ethical concerns, primarily related to user agency. Because recommender systems control the information users see to a large extent, they have the potential to influence their ideologies and behavior in the long run. Users can get caught in filter bubbles that only serve to reinforce their biases and insulate them from opposing viewpoints. Additionally, users may be vulnerable to their recommendations being manipulated by third parties. For instance, a third party looking to propagate their viewpoint might coordinate their interactions with the recommender system in a manner that induces certain items/posts to be rank highly in the recommendation lists of some target set of users. As detailed in the paper, both these scenarios imply a lack of user agency and the metrics we propose quantify this so that these issues can potentially be identified.

**Reproducibility:** We link a rough version of the code we use for our experiments as supplementary material and additionally detail the algorithms we implement in Appendix C.

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

# A    STRUCTURAL CAUSAL MODELS

In this work, we use the SCM framework (Pearl, 2009) to model causal relationships. An SCM is a tuple $\mathcal{M} = (\mathbf{U}, \mathbf{V}, \mathcal{F}, \mathbb{P}(\mathbf{U}))$, where $\mathbf{U}$ and $\mathbf{V}$ are the sets of exogenous and endogenous nodes. An SCM implies a directed acyclic graph (DAG) $\mathcal{G}$ over the nodes $\mathbf{V}$ with $\mathrm{Pa}_i$ denoting the parents of node $V_i \in \mathbf{V}$. Each $V_i \in \mathbf{V}$ is generated using the structural equation $V_i := f_i(\mathrm{Pa}_i, U_i)$, where $\mathrm{Pa}_i \subset \mathbf{V}$, $U_i \in \mathbf{U}$, and $f_i \in \mathcal{F}$. The joint distribution $\mathbb{P}(\mathbf{U})$ induces the observational distribution $\mathbb{P}(\mathbf{V})$. An interventional distribution is generated by the (modified) SCM $\mathcal{M}_x = (\mathbf{U}, \mathbf{V}, \mathcal{F}_x, \mathbb{P}(\mathbf{U}))$, where $\mathcal{F}_x$ is the set of functions obtained by replacing the structural equation for $X \in \mathbf{V}$ with $X := x$. Informally, it represents the state of the system after intervening on $X$ and setting it to $x$, denoted by $\mathrm{do}(X = x)$. SCMs also allow us to compute counterfactual quantities that express what would have happened, had we set the node $X$ to $x$, given that the (factual) event $E$ occurred. Counterfactuals distributions are generated using the SCM $\mathcal{M}_{\mathrm{cf}} := (\mathbf{U}, \mathbf{V}, \mathcal{F}_x, \mathbb{P}(\mathbf{U}|E))$.

# B    DETAILS FOR SECTION 5

## B.1    SCORE-BASED RECOMMENDER SYSTEMS

Given data of $T$ time steps,

$$\min_{f \in \mathcal{F}, g \in \mathcal{G}} \sum_{t \in [T]} \sum_{i \in [n]} (\mathbf{o}_{i,t}[j] - f(\mathbf{h}_{i,t})^\top g(\mathbf{h}_{j,t}^v))^2 + \Omega(f, g), \tag{6}$$

where $\mathbf{h}_{j,t}^v$ is the rating history for item $j$ so far. In matrix factorization, the user representation $f(\mathbf{h}_{i,t}) = \mathbf{p}_i \in \mathbb{R}^d$ and item representation $g(\mathbf{h}_{j,t}^v) = \mathbf{p}_j \in \mathbb{R}^d$ are learnable constant vectors. Another setting could be that $f, g$ are both neural networks, e.g., LSTMs, that map user and item history to a $d$-dimensional vector. In both cases, the regularizer $\Omega(f, g)$ are the norms of the user and item vectors. Once a score $\hat{s}_{i,j,t} = \hat{f}(\mathbf{h}_{i,t})^\top \hat{g}(\mathbf{h}_{j,t}^v)$ is learned using the dataset $\mathbf{d}_t$, the recommendation probability $\mathbb{P}(\mathbf{A}_{i,t} = j | \mathrm{do}(\mathbf{D}_t = \mathbf{d}_t))$ for example can be a pointmass $\mathbb{I}\{j = \arg\max_{j \in \mathcal{V}} \hat{s}_{i,j,t}\}$, or follows a softmax policy $e^{\beta \hat{s}_{i,j,t}}/(\sum_l e^{\beta \hat{s}_{i,l,t}})$ for $\beta \geq 0$ controlling how stochastic one wants the recommender to be.

## B.2    PROOFS

For the proofs, we consider a matrix factorization model with learned user factors given by $P \in \mathbb{R}^{n \times d}$ and learned item factors given by $Q \in \mathbb{R}^{m \times d}$.

*Proof of Proposition 5.1.* For a fixed factual recommendation trajectory $\mathbf{h}_{i,t-k:}^\star = (\mathbf{a}_{i,t-k}^\star, \mathbf{o}_{i,t-k}^\star, \cdots, \mathbf{a}_{i,t-1}^\star, \mathbf{o}_{i,t-1}^\star)$, the optimization problem in equation 2 is

$$\max_{\mathbf{o}_{1:k} \in \mathbb{R}^k} \mathbb{P}(\mathbf{A}_{i,t} = j | \mathbf{A}_{i,t-k}^\star = \mathbf{a}_{i,t-k}^\star, \mathrm{do}(\mathbf{O}_{i,t-k} = \mathbf{o}_1), \ldots, \mathbf{A}_{i,t-1}^\star = \mathbf{a}_{i,t-1}^\star, \mathrm{do}(\mathbf{O}_{i,t-1} = \mathbf{o}_k)),$$

In this setting, the user $i$ modifies the ratings $\mathbf{o}_{1:k}$ for the items in the factual trajectory $\mathbf{a}_{i,t-k:t-1}^\star$. Since retraining the entire recommender system is not feasible after every user interaction, we make the following simplifying assumption for how the user vector is updated after a rating $o_k$ is modified.

**Assumption**: After each rating user $i$ modifies, the user vector $P_i$ (the $i^{\text{th}}$ row of $P$) is updated but $Q$ is kept unchanged. The objective of matrix factorization is to solve following expression, where $R$ is the user-item rating matrix:

$$\min_{P,Q} \|PQ^T - R\|_2^2$$

Under this assumption, the updated user vector after every interaction is given by:

$$P_i = \arg\min_{p'} \sum_{v \in V} (p'^T Q_v - R_{iv})^2$$

where $V$ is the subset of items the user $i$ has interacted with. Every interaction adds an additional element to $V$ and populates $R_{iv}$. After $k$ timesteps(from $t - k$ to $t - 1$), this optimization problem has a simple closed-form solution given by:

$$P_i = (Q_{\mathrm{rated}}^T Q_{\mathrm{rated}})^{-1} Q_{\mathrm{rated}}^T R_{\mathrm{rated}},$$

where

$$
Q_{\text{rated}} = \begin{bmatrix} Q_{\mathbf{a}^\star_{i,1}} \\ Q_{\mathbf{a}^\star_{i,2}} \\ \vdots \\ Q_{\mathbf{a}^\star_{i,t-1}} \end{bmatrix}, \quad R_{\text{rated}} = \begin{bmatrix} R_{i,1} \\ \vdots \\ R_{i,t-k-1} \\ \mathbf{o}_1 \\ \mathbf{o}_3 \\ \vdots \\ \mathbf{o}_k \end{bmatrix}.
$$

$Q_{\text{rated}} \in \mathbb{R}^{(t-1)\times d}$ is a matrix whose rows are the item vectors of the items rated by the user upto time $t$ and $R_{\text{updated}}$ is a column vector of size $t_0 - 1$ which represents all the corresponding ratings given to these items by the user $i$ up to time $t$ with the last $k$ ratings being replaced by $\mathbf{o}_1, \mathbf{o}_2, \ldots \mathbf{o}_k$ as discussed above.

We assume a stochastic $\beta$-softmax selection rule given by:

$$
\mathbb{P}(\mathbf{A}_{i,t_0} = j) = \frac{e^{\beta s_{ij}}}{\sum_{k\in V} e^{\beta s_{ik}}}
$$

where $s_{ij} = P_i^T Q_j$ is the predicted rating for the interaction between user $i$ and item $j$. In this case,

$$
\mathbb{P}(\mathbf{A}_{i,t} = j | \mathbf{A}^\star_{i,t-k} = \mathbf{a}^\star_{i,t-k}, \text{do}(\mathbf{O}_{i,t-k} = \mathbf{o}_1), \ldots, \mathbf{A}^\star_{i,t-1} = \mathbf{a}^\star_{i,t-1}, \text{do}(\mathbf{O}_{i,t-1} = \mathbf{o}_k)) = \frac{e^{\beta P_i^T Q_j}}{\sum_{k\in V} e^{\beta P_i^T Q_k}}
$$

Maximizing this is equivalent to maximizing the logarithm of this quantity, given by:

$$
\beta P_i^T Q_j - \underset{k \in V}{\text{LSE}}(\beta P_i^T Q_k)
$$

where LSE denotes the log-sum-exponential. We can rewrite the objective as:

$$
\max_{\mathbf{o}_{1,\ldots k}\in\mathbb{R}^k} \beta P_i^T Q_j - \underset{k \in V}{\text{LSE}}(\beta P_i^T Q_k) = \min_{\mathbf{o}_{1,\ldots k}\in\mathbb{R}^k} -\beta P_i^T Q_j + \underset{k \in V}{\text{LSE}}(\beta P_i^T Q_k)
$$

We can see that $P_i^T Q_j$ is a linear function in all $\mathbf{o}_1, \ldots \mathbf{o}_k$ since every $Q_j$ is independent of $\mathbf{o}_1, \mathbf{o}_2 \ldots \mathbf{o}_k$.

Let $\mathbf{o} = [\mathbf{o}_1, \ldots \mathbf{o}_k]^T$. Then, for some $b_k \in \mathbb{R}^t$ and $c_k \in \mathbb{R} \forall k \in V$, the objective becomes:

$$
\min_{\mathbf{o}_{1,\ldots k}\in\mathbb{R}^k} \underset{k \in V}{\text{LSE}} \left(\beta(b_k^T \mathbf{o} + c_k)\right) - \beta(b_j^T \mathbf{o} + c_j)
$$

This is convex because log-sum-exp is a convex function, affine functions are convex, and the composition of a convex and an affine function is convex. $\qquad\square$

**Lemma B.1.** *Let $f : \mathbb{R}^n \to \mathbb{R}$ be a convex function with only one root, i.e., $f(\mathbf{x}_0) = 0$ for some $\mathbf{x}_0 \in \mathbb{R}^n$. Then, $g(\mathbf{x}) = [f(\mathbf{x})]^2$ is a quasi-convex function. **Proof:** To prove that $g(\mathbf{x})$ is quasi-convex, we need to show that for any $\alpha \in \mathbb{R}$, the sublevel set $S_\alpha = \{\mathbf{x} \in \mathbb{R}^n | g(\mathbf{x}) \leq \alpha\}$ is a convex set.*

1. *If $\alpha < 0$, the sublevel set $S_\alpha$ is empty because $[f(\mathbf{x})]^2 \geq 0$ for all $\mathbf{x} \in \mathbb{R}^n$. An empty set is convex by definition.*

2. *If $\alpha = 0$, the sublevel set $S_0 = \{\mathbf{x}_0\}$ because $f(\mathbf{x}_0) = 0$ and $[f(\mathbf{x})]^2 > 0$ for all $\mathbf{x} \neq \mathbf{x}_0$. A singleton set is convex.*

3. *If $\alpha > 0$, consider the sublevel set $S_\alpha = \{\mathbf{x} \in \mathbb{R}^n | [f(\mathbf{x})]^2 \leq \alpha\}$. We can rewrite this as:*

$$
S_\alpha = \{\mathbf{x} \in \mathbb{R}^n | -\sqrt{\alpha} \leq f(\mathbf{x}) \leq \sqrt{\alpha}\},
$$

*which is a convex set for any convex $f$.*

*Proof of Proposition 5.2.* For a fixed factual recommendation trajectory $\mathbf{h}^{\star}_{i_2,t-k:} = (\mathbf{a}^{\star}_{i_2,t-k}, \mathbf{o}^{\star}_{i_2,t-k}, \cdots,$
$\mathbf{a}^{\star}_{i_2,t-1}, \mathbf{o}^{\star}_{i_2,t-1})$, the optimization problem in equation 4 can be written as:

$$\max_{\mathbf{o}_{1,\ldots k}\in\mathbb{R}^k} d(\mathbb{P}(\mathbf{A}_{i_1,t}|\mathbf{A}^{\star}_{i_2,t-k} = \mathbf{a}^{\star}_{i_2,t-k}, \mathrm{do}(\mathbf{O}_{i_2,t-k} = \mathbf{o}_1)),\ldots,\mathbf{A}^{\star}_{i_2,t-1} = \mathbf{a}^{\star}_{i_2,t-1}, \mathrm{do}(\mathbf{O}_{i_2,t-1} = \mathbf{o}_k)), \mathbb{P}(\mathbf{A}_{i_1,t}))$$

In this setting, the user $i_2$ modifies the ratings of the factual item trajectory $\mathbf{a}^{\star}_{i_2,t-k:t-k+1}$. Since retraining the entire recommender system is not feasible after every user interaction, we make the following simplifying assumption for how the item vector is updated after a rating $o_k$ is modified.

**Assumption**: After $i_2$ rates an item $j$, the item vector $Q_j$(the $j^{\text{th}}$ row of $Q$) is updated but $P$ is kept unchanged. Similar to the proof past-$k$ reachability, the updated item vector for an item $j$ rated by $i_2$ is given by:

$$Q_j = (P^T_{\text{rated}}P_{\text{rated}})^{-1}P^T_{\text{rated}}R_{\text{rated}},$$

where

$$P_{\text{rated}} = \begin{bmatrix} P_{\mu^*_{j,1}} \\ \vdots \\ P_{i_2} \\ \vdots \\ P_{\mu^*_{j,t-1}} \end{bmatrix}, \quad R_{\text{rated}} = \begin{bmatrix} R_{j,1} \\ \vdots \\ \mathbf{o}_t \\ \vdots \\ R_{j,t-1} \end{bmatrix}.$$

Here, $P_{\text{rated}}$ is a matrix $\in \mathbb{R}^{(t-1)\times d}$ whose rows are the user vectors of the users that rated the item $j$ up to $t$, that is represented by $\mu^*_{j,t}$; with user $i_2$ rating it at $t-k+t-1$ and $R_{\text{rated}}$ is a column vector of size $t-1$ that represents all the corresponding ratings given by these users to $j$, with the rating at time $t+k'-1$ being replaced with $\mathbf{o}_{k'+k}$.

We assume a stochastic $\beta$-softmax selection rule given by:

$$\mathbb{P}(\mathbf{A}_{i,t_0} = j) = \frac{e^{\beta s_{ij}}}{\sum_{k\in V} e^{\beta s_{ik}}}$$

where $s_{ij} = P^T_i Q_j$ is the predicted rating for the interaction between user $i$ and item $j$.

The $L_2$ distance metric is defined as the defined as the $L_2$ distance between the discrete probability probability distributions of $i_1$'s next recommended item before and after $i_2$'s modifies their ratings.

$$f(\mathbf{o}) = \sum_{j\in V} \Bigg(\mathbb{P}\Big(\mathbf{A}_{i_1,t_0} = j \mid \mathbf{A}^{\star}_{i_2,t_0-k} = \mathbf{a}^{\star}_{i_2,t_0-k}, \mathrm{do}\left(\mathbf{O}_{i_2,t_0-k} = \mathbf{o}_1\right),\ldots,\mathbf{A}^{\star}_{i_2,t_0-1} = \mathbf{a}^{\star}_{i_2,t_0-1},$$

$$\mathrm{do}\left(\mathbf{O}_{i_2,t_0-1} = \mathbf{o}_k\right)\Big) - \mathbb{P}\left(\mathbf{A}_{i_1,t_0} = j\right)\Bigg)^2$$

We note that the predicted rating of a user-item interaction only changes when the item is one of the items for which the user $i_2$ modifies their ratings. In this case, we can see that the predicted rating given by $i_1$ on an item $i_2$ rates $\mathbf{o}_k$ is a linear function in $\mathbf{o}_k$. Our objective can be written as $f(\mathbf{o}) = \sum_{j\in V} g_j(\mathbf{o})$, where

$$g_j(\mathbf{o}) = \left(\frac{e^{\beta(c_{i_1 j}\mathbf{o}_k + d_{i_1 j})}}{\sum_{k\in V} e^{\beta(c_{i_1 k}\mathbf{o}_k + d_{i_1 k})}} - \mathbb{P}(\mathbf{A}_{i_1,t_0} = j)\right)^2$$

We can rewrite this as:

$$g_j(r) = \left(\frac{e^{\beta(c_{i_1 j}\mathbf{o}_k + d_{i_1 j})}}{\sum_{k\in V} e^{\beta(c_{i_1 k}\mathbf{o}_k + d_{i_1 k})}} - C_j\right)^2$$

where $C_j = \mathbb{P}\left(\mathbf{A}_{i_1,t_0} = j\right)$ is a constant. Now, let's consider the function:

$$h_j(\mathbf{o}) = \frac{e^{\beta(c_{i_1 j}\mathbf{o}_k + d_{i_1 j})}}{\sum_{k\in V} e^{\beta(c_{i_1 k}\mathbf{o}_k + d_{i_1 k})}}$$

This function is the softmax function, which is known to be convex (Boyd and Vandenberghe, 2004). Next, consider the function:

$$l_j(\mathbf{o}) = h_j(\mathbf{o}) - C_j$$

The difference of a convex function and a constant is also convex (Boyd and Vandenberghe, 2004). Therefore, $l_j(\mathbf{o})$ is convex. Finally, let's look at:

$$g_j(\mathbf{o}) = (l_j(\mathbf{o}))^2$$

Each $l_j$ is a monotonic convex function with exactly one root. By Lemma B.1, each $g_j(\mathbf{o})$ is quasiconvex. Now, we can write the stability objective function as:

$$f(\mathbf{o}) = \sum_{j \in V} g_j(\mathbf{o})$$

The sum of quasiconvex functions is also quasiconvexconvex (Boyd and Vandenberghe, 2004). Since $f(r)$ is quasiconvex, in practice, if we optimize each $o_k$ in the interval $[a, b] \subset \mathbb{R}$, the maximum value of $f(r)$ is attained at an extreme point of $[a, b]$ (Boyd and Vandenberghe, 2004). In other words, the stability objective is quasiconvex and achieves its optimal value at a boundary point of the domain. $\qquad\square$

## C  ALGORITHM DETAILS

- **Past Reachability:** Details provided in 1. We use a model trained up to time $t_0 - k$, and then aim to optimize the user's ratings for the factual next $k$ items with the objective of maximizing the probability of recommending to item to be reached after $k$ steps, with a stochastic recommendation policy given by:

$$\mathbb{P}(\mathbf{A}_{i,t_0} = j) = \frac{e^{\beta s_{ij}}}{\sum_{k \in V} e^{\beta s_{ik}}}$$

  where $s_{ij}$ is the predicted rating for the interaction between user $i$ and item $j$. This is a stochastic selection rule controlled by a parameter of stochasticity $\beta$ to select items after the recommender system ranks them. There are $k$ parameters of optimization, where $k$ is the time horizon, corresponding to the user's action at each time step.

- **Future Reachability:** Details provided in 2. We use a model trained up to time $t_0$ and the user follows the recommendations given by a deterministic recommender system that returns the top-1 item that the user hasn't already rated based on predicted score, with the same objective mentioned above. There are $k \times |V|$ parameters of optimization here, where $|V|$ is the size of the item vocabulary. The $(k \cdot m + t)$th parameter represents the user's action if they were to be recommended item $m$ at time $t_0 + t$.

- **Past Stability:** Details provided in 3. We use a model trained up to time $t_0 - k$, and then aim to optimize the adversary's ratings for the factual next $k$ items with the objective of maximally changing the user's recommended list, with the same stochastic recommendation policy mentioned above. We use Hellinger Distance as our distance metric. There are $k$ parameters of optimization, where $k$ is the time horizon, corresponding to the adversary's action at each time step.

- **Future Stability:** Details provided in 4. We use a model trained up to time $t_0$ and the adversary follows the recommendations given by a deterministic recommender system that returns the top-1 item that the user hasn't already rated based on predicted score, with the same objective mentioned above. There are $k \times |V|$ parameters of optimization here, where $|V|$ is the size of the item vocabulary. The $(k \cdot m + t)$th parameter represents the adversary's action if they were to be recommended item $m$ at time $t_0 + t$.

Note: We use a deterministic selection rule for computing future based metrics because it does not suffer from too much variation across epochs, unlike the stochastic selection rule, which would require an even larger parameter space to account for every possible item sequence.

## D  EXPERIMENTAL DETAILS

**Dataset Summary Statistics**  The table below shows some dataset summary statistics.

---

**Algorithm 1:** past-$k$ reachability

---

**Input:** User $i$, Item to be reached $j$, Time Horizon $k$
**Output:** Optimal ratings for history items
1 initialize chosen ratings $\mathbf{o}$ for history items as their factual ratings; **for** *epoch* $\leftarrow 1$ **to** $n_{epochs}$ **do**
2     $\mathbf{o}$ clamped between [1,5]
3     **for** *timestep* $\leftarrow 1$ **to** $k$ **do**
4        next item $m \leftarrow$ historical item at $t_0 - t + timestep$
5        Update user vector based on (item, rating) pair $(m, \mathbf{o}_{timestep})$
6     Compute $\mathbb{P}(\mathbf{A}_{i,t_0} = j)$
7     Backpropagate to chosen ratings $\mathbf{o}$

---

**Algorithm 2:** future-$k$ reachability

---

**Input:** User $i$, Item to be reached $j$, Time Horizon $k$
**Output:** Optimal ratings for future items
1 initialize parameter space $R$ of size $k \cdot |V|$ randomly; **for** *epoch* $\leftarrow 1$ **to** $n_{epochs}$ **do**
2     $\mathbf{o}$ clamped between [1,5]
3     initialize reachability_vals of size num_samples
4     **for** *avg* $\leftarrow 1$ **to** $n_{num\_samples}$ **do**
5        **for** *timestep* $\leftarrow 1$ **to** $k$ **do**
6           next item $m \leftarrow$ top-1 item in recommended list
7           Update user vector based on (item, rating) pair $(m, R_{k \cdot m + timestep})$
8        reachability_vals$_{avg} \leftarrow \mathbb{P}(\mathbf{A}_{i,t_0} = j)$
9     Compute mean of reachability_vals
10     Backpropagate to parameter space $R$

---

**Algorithm 3:** past-$k$ stability

---

**Input:** User $i_1$, Adversary $i_2$, Time Horizon $k$
**Output:** Optimal ratings for history items
1 initialize chosen ratings $\mathbf{o}$ for history items as their factual ratings;
2 $i_1$'s initial preferences $l_1 \leftarrow recsys(i_1,$ item vectors$)$
3 **for** *epoch* $\leftarrow 1$ **to** $n_{epochs}$ **do**
4     $\mathbf{o}$ clamped between [1,5]
5     **for** *timestep* $\leftarrow 1$ **to** $k$ **do**
6        next item $m \leftarrow$ historical item at $t_0 - t + timestep$
7        Update item vector for $m$ based on (user, rating) pair $(i_2, \mathbf{o}_{timestep})$
8     $i_1$'s final preferences $l_2 \leftarrow recsys(i_1,$ item vectors$)$
9     Compute distance $d(l_1, l_2)$
10     Backpropagate this to chosen ratings $\mathbf{o}$

---

---

**Algorithm 4:** future-$k$ stability

---

**Input:** User $i_1$, Adversary $i_2$, Time Horizon $k$
**Output:** Optimal ratings for future items
1   initialize parameter space $R$ of size $k \cdot |V|$ randomly
2   $i_1$'s initial preferences $l_1 \leftarrow recsys(i_1,$ item vectors$)$
3   **for** *epoch* $\leftarrow 1$ **to** $n_{epochs}$ **do**
4      |   **o** clamped between [1,5]
5      |   initialize stability_vals of size num_samples
6      |   **for** *avg* $\leftarrow 1$ **to** $n_{num\_samples}$ **do**
7      |     |   **for** *timestep* $\leftarrow 1$ **to** $t$ **do**
8      |     |     |   next item $m \leftarrow$ top-1 item in recommended list
9      |     |     |   Update item vector for $m$ based on (user, rating) pair $(i_2, R_{k \cdot m + timestep})$
10     |     |   $i_1$'s final preferences $l_2 \leftarrow recsys($user vector, item vectors$)$
11     |     |   stability_vals$_{avg} \leftarrow d(l_1, l_2)$, d=distance metric
12     |   Compute mean of stability_vals
13     |   Backpropagate this to parameter space $R$

---

| Data set | ML 1M |
|---|---|
| **Users** | 6040 |
| **Items** | 3706 |
| **Ratings** | 1000209 |
| **Density (%)** | 4.47% |

Table I: Statistics and performance metrics for the ML 1M dataset.

**Additional Recommender System Details:**    We choose 100 as the size of both user and item latent vectors for matrix factorization.

For the Recurrent Recommender Network, we train two LSTMs in parallel: one that takes a user's item history as input and outputs a user vector and another that takes an item's user history as input and outputs an item vector.

The user's item history is a list of the form [(item_id$_k$, rating$_k$)] where $k$ varies from 1 to $n$, if $n$ is the number of interactions the user has had upto this point.

Each LSTM has input size 2 and hidden size 100, and we choose the last hidden state as the user/item vector.

Similar to general factorization based recommenders, the rating for a particular interaction is given by the dot product of the user and item vector for the (user,item) pair involved in the interaction.

For training, we first sort every interaction in the dataset by timestep and attempt to predict the rating for the next interaction based on interactions that have already taken place.

We train three versions of each recommender system: one with complete data, one with the last item for each user left out and one with the last 5 items rated by each user left out in order to run our experiments.

**Updating the recommender system after every interaction:** We acknowledge that re-training the recommender system after every new user-item interaction is not feasible and is not done in practice in commercial systems(Yao et al., 2021). Both matrix factorization and recurrent recommender networks are inherently factorization based and create latent user and item representations that are updated whenever the model retrains. Instead of retraining the model, for computing reachability based metrics, we operate under the assumption that the item latent vectors remain fixed while we update the user latent vectors by solving a closed form equation given in 7 for matrix factorization. Similarly, for computing stability based metrics, we operate under the assumption that the user latent vectors remain fixed while we update the item latent vectors by solving a closed form equation given

in 8 for matrix factorization. We borrow this approach from (Yao et al., 2021).

$$p = \underset{p'}{\mathrm{argmin}} \sum_{v \in \mathcal{V}_{\text{seen}}} \left( p'^T q_v - r_v \right)^2 \tag{7}$$

$$q = \underset{q'}{\mathrm{argmin}} \sum_{u \in \mathcal{U}_{\text{seen}}} \left( p_u^T q' - r_u \right)^2 \tag{8}$$

Here, $p$ denotes the user latent vector and $q$ denotes the item latent vector. $\mathcal{V}_{\text{seen}}$ is the list of items the user has rated, $r_v$ is the rating given to item $v$. Similarly, $\mathcal{U}_{\text{seen}}$ is the list of users the item has been rated by and $r_v$ is the rating given by user $u$.

For the recurrent recommender network, this is even more straightforward as we leave the LSTM as is and simply query it with the edited user/item history to obtain new user/item vectors

**Settings for plots:** Regarding the plots in 6,

- For reachability based metrics, we randomly choose 10% of users as set $A$ and 10% of items as set $B$ and compute the reachability of every (user,item) pair in $A \times B$.
- For stability based metrics, we randomly choose 20% of users and divide them equally among set $A$ and set $B$ to serve as the primary user set and adversary set respectively. We then compute the stability of every combination of (user,adversary) pair in $A \times B$.

**Compute Requirements**   We make use of 40G A6000 GPUs.

# E   METRICS FOR AGGREGATED GROUPS

As an additional experiment, we group items by their popularity and attempt to see if more popular items are more reachable than less popular ones.

For this, we use the number of interactions an item is a part of in the training set as a proxy for it's popularity among users. We consider two groups of items:

- Set A consists of the 30 most popular items.
- Set B consists of items with intermediate popularity (30 items between 200th to 300th in popularity).

We then randomly choose 10% of the total users as set C and compute the future reachability for every (user,item) pair in $C \times A$ and compare it to the future reachability of every (user,item) pair in $C \times B$.

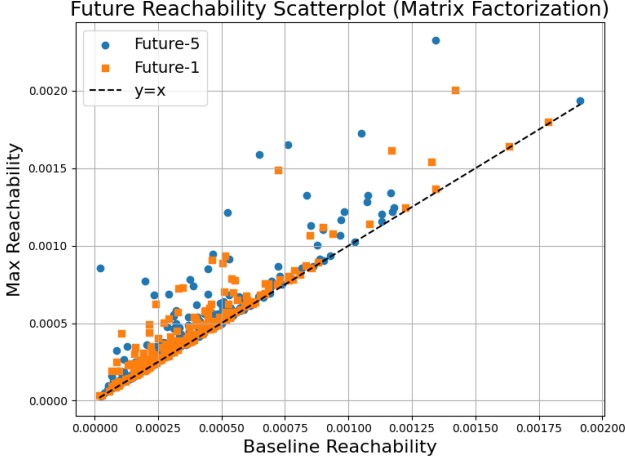

Figure 6: Future Reachability(MF) for popular items vs items with intermediate popularity

| Item Type | Popular item | Intermediate Item |
|---|---|---|
| Mean Lift | 1.3735 | 1.1846 |
| Lower Confidence Interval | 1.0500 | 1.1361 |
| Upper Confidence Interval | 1.6971 | 1.2330 |

Table II: Future Reachability with Popular vs Intermediate Items

**Observations:** The baseline values for future reachability for popular and intermediate items were observed to be similar on average. The results in Table II then show us that popular items have higher values of max future reachability on average as compared to items with intermediate popularity. We note that this difference is not as prominent when measuring past reachability instead of future reachability.

We also group users by their activity and attempt to see if active users act as better adversaries on average than inactive users, i.e., whether active users can cause a larger change in a random user's recommendation list than inactive users.

Using the number of ratings given by users as a proxy for their activity, we consider two groups of users:

- Set A consists of the 30 most active users.
- Set B consists of users with intermediate activity (30 users between 200th to 300th in activity).

We then randomly choose 10% of the remaining users as set C, to act as primary users and compute the stability for every (user,adversary) pair in $C \times A$ and compare it to the reachability of every (user,item) pair in $C \times B$. Figure 7 shows the resulting histogram for future instability.

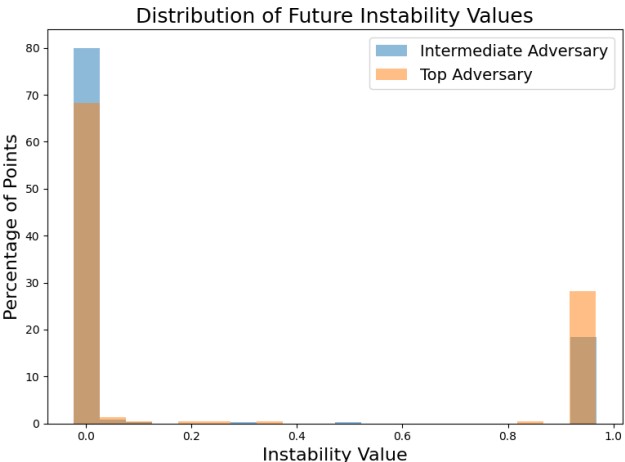

Figure 7: Future Instability(MF) for Active vs Intermediate users

| Adversary | Active User | Intermediate User |
|---|---|---|
| Mean | 0.2962 | 0.1938 |
| Lower Confidence Interval | 0.2395 | 0.1549 |
| Upper Confidence Interval | 0.3527 | 0.2327 |

Table III: Future Instability with Active vs Intermediate Adversaries

**Observations:** These results show us that active users introduce more instability on average as compared to users with intermediate levels of activity. We note that this effect is not as prominent when measuring past stability instead of future stability.

We perform both these experiments on the Matrix Factorization based recommender system.

# F  SELECTED FIGURES WITH ADDITIONAL PARAMETER VALUES

**1) Scatterplot for reachability with additional time horizons, for comparison $\beta = 0.8$**

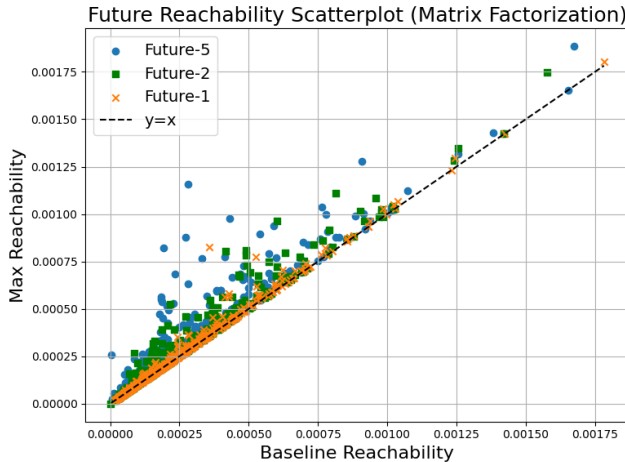

Figure 8: Future Reachability: Time horizon = 1 vs Time horizon = 2 vs Time horizon = 5

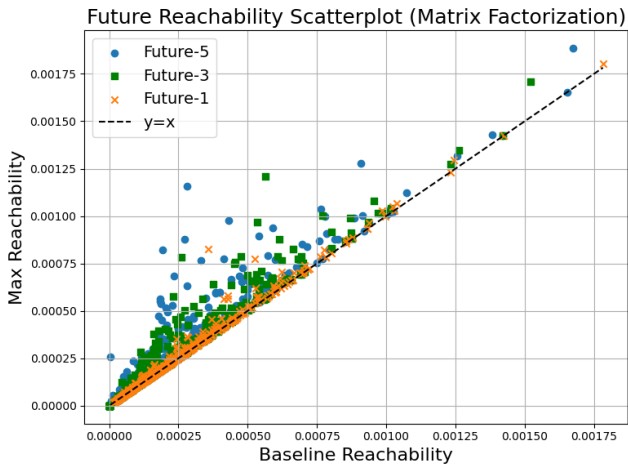

Figure 9: Future Reachability: Time horizon = 1 vs Time horizon = 3 vs Time horizon = 5

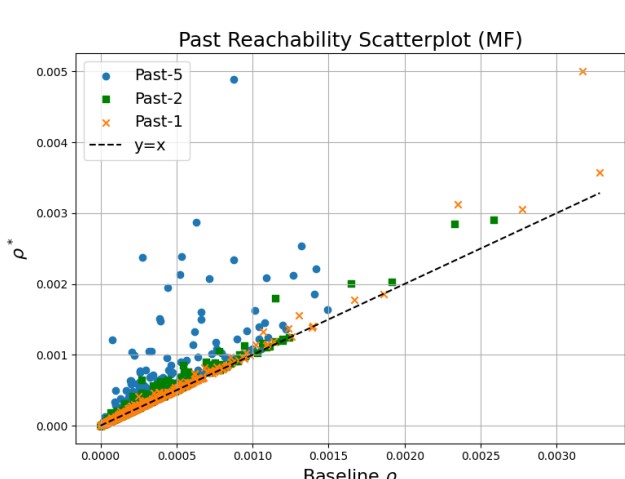

Figure 10: Past Reachability: Time horizon = 1 vs Time horizon = 2 vs Time horizon = 5

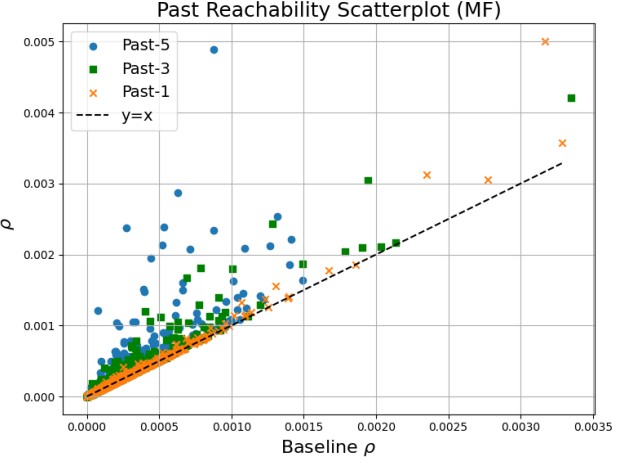

Figure 11: Past Reachability: Time horizon = 1 vs Time horizon = 3 vs Time horizon = 5

**2) Scatterplots for past stability with different stochasticities**

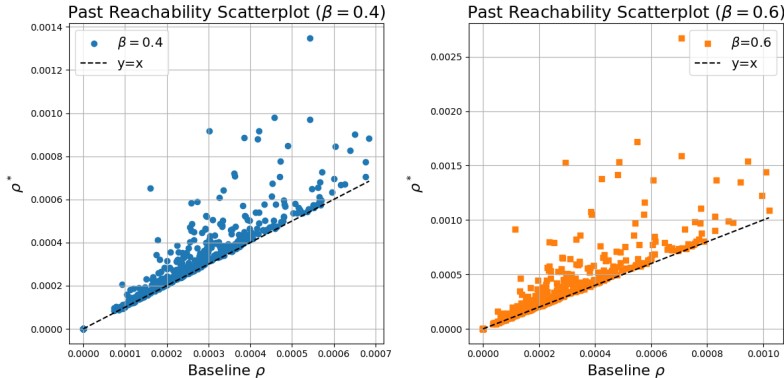

Figure 12: $\beta = 0.4$ vs $\beta = 0.8$

**3) Mean Past Instability as a function of time horizon**

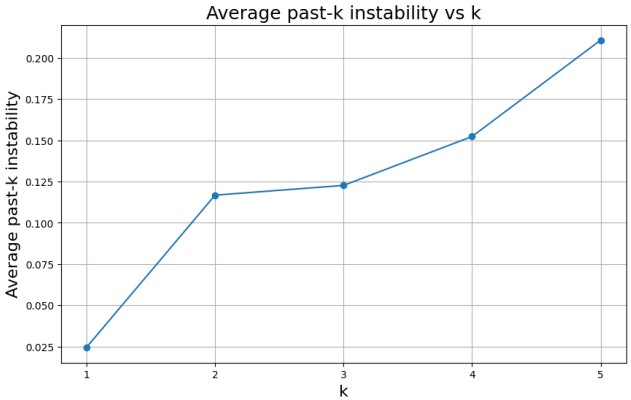

Figure 13: Mean Past Instability vs Time Horizon

**4) Past stability for different time horizons** $\beta = 0.8$

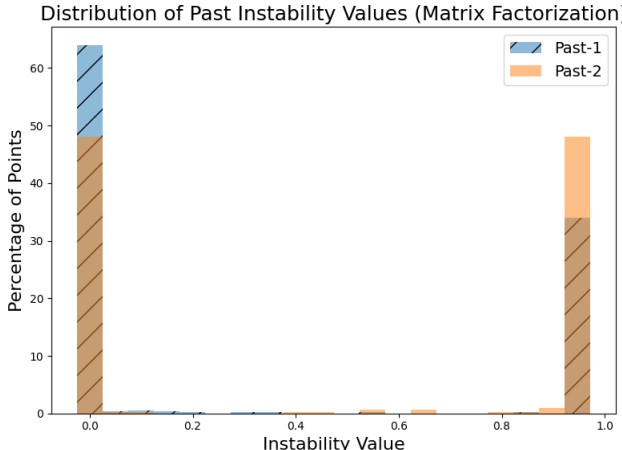

Figure 14: Time horizon=1 vs Time Horizon=2

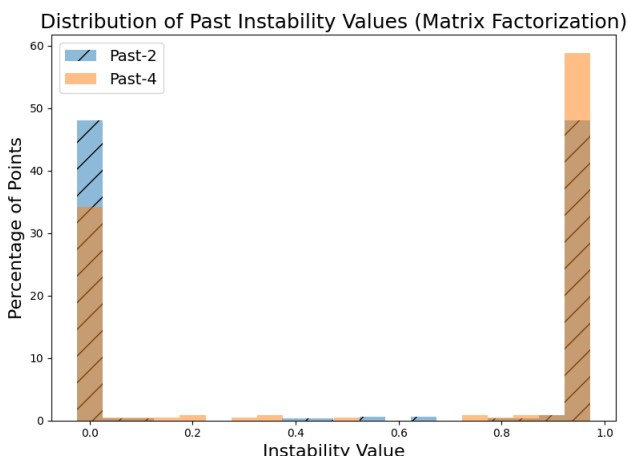

Figure 15: Time Horizon=2 vs Time Horizon=4

# G   GRADIENT EXPRESSIONS

In this section, we write out the gradient expressions for both reachability and stability for multiple timesteps to give a better idea of the gradients one might need access to in a whitebox setting.

## G.1   REACHABILITY GRADIENT

### G.1.1   2 TIMESTEPS

• Gradient with respect to $f_1$

$$\nabla_{f_1} \mathbb{E}_{A_{i,t}, A_{i,t+1}} \left[ P(A_{i,t+2} = j | \mathrm{do}(O_{i,t} = f_1(A_{i,t})), \mathrm{do}(O_{i,t+1} = f_2(A_{i,t+1}))) \right]$$
$$= \sum_{a_{i,t}} P(A_{i,t} = a_{i,t}) \nabla_{f_1} \sum_{a_{i,t+1}} \left[ P(A_{i,t+1} = a_{i,t+1} | \mathrm{do}(O_{i,t} = f_1(a_{i,t}))) \right.$$
$$\left. P(A_{i,t+2} = j | \mathrm{do}(O_{i,t+1} = f_2(a_{i,t+1})), \mathrm{do}(O_{i,t} = f_1(a_{i,t}))) \right]$$

• Gradient with respect to $f_2$

$$\nabla_{f_2} \mathbb{E}_{A_{i,t}, A_{i,t+1}} \left[ P(A_{i,t+2} = j | \mathrm{do}(O_{i,t} = f_1(A_{i,t})), \mathrm{do}(O_{i,t+1} = f_2(A_{i,t+1}))) \right]$$
$$= \sum_{a_{i,t}} P(A_{i,t} = a_{i,t}) \sum_{a_{i,t+1}} P(A_{i,t+1} = a_{i,t+1} | \mathrm{do}(O_{i,t} = f_1(a_{i,t})))$$
$$\nabla_{f_2} P(A_{i,t+2} = j | \mathrm{do}(O_{i,t+1} = f_2(a_{i,t+1})), \mathrm{do}(O_{i,t} = f_1(a_{i,t})))$$

### G.1.2   3 TIMESTEPS

• Gradient with respect to $f_1$

$$\nabla_{f_1} \mathbb{E}_{A_{i,t}, A_{i,t+1}, A_{i,t+2}} [P(A_{i,t+3} = j | \mathrm{do}(O_{i,t} = f_1(A_{i,t})),$$
$$\mathrm{do}(O_{i,t+1} = f_2(A_{i,t+1})), \mathrm{do}(O_{i,t+2} = f_3(A_{i,t+2})))]$$
$$= \sum_{a_{i,t}} P(A_{i,t} = a_{i,t}) \nabla_{f_1} \sum_{a_{i,t+1}} [P(A_{i,t+1} = a_{i,t+1} | \mathrm{do}(O_{i,t} = f_1(a_{i,t})))$$
$$\sum_{a_{i,t+2}} P(A_{i,t+2} = a_{i,t+2} | \mathrm{do}(O_{i,t+1} = f_2(a_{i,t+1})), \mathrm{do}(O_{i,t} = f_1(a_{i,t})))$$
$$P(A_{i,t+3} = j | \mathrm{do}(O_{i,t+2} = f_3(a_{i,t+2})),$$
$$\mathrm{do}(O_{i,t+1} = f_2(a_{i,t+1})), \mathrm{do}(O_{i,t} = f_1(a_{i,t})))]$$

- Gradient with respect to $f_2$

$$\nabla_{f_2}\mathbb{E}_{A_{i,t},A_{i,t+1},A_{i,t+2}}\left[P(A_{i,t+3}=j|\text{do}(O_{i,t}=f_1(A_{i,t})),\text{do}(O_{i,t+1}=f_2(A_{i,t+1})),\text{do}(O_{i,t+2}=f_3(A_{i,t+2})))\right]$$

$$=\sum_{a_{i,t}}P(A_{i,t}=a_{i,t})\sum_{a_{i,t+1}}P(A_{i,t+1}=a_{i,t+1}|\text{do}(O_{i,t}=f_1(a_{i,t})))$$

$$\nabla_{f_2}\sum_{a_{i,t+2}}[P(A_{i,t+2}=a_{i,t+2}|\text{do}(O_{i,t+1}=f_2(a_{i,t+1})),\text{do}(O_{i,t}=f_1(a_{i,t})))$$

$$P(A_{i,t+3}=j|\text{do}(O_{i,t+2}=f_3(a_{i,t+2})),\text{do}(O_{i,t+1}=f_2(a_{i,t+1})),\text{do}(O_{i,t}=f_1(a_{i,t})))]$$

- Gradient with respect to $f_3$

$$\nabla_{f_3}\mathbb{E}_{A_{i,t},A_{i,t+1},A_{i,t+2}}\left[P(A_{i,t+3}=j|\text{do}(O_{i,t}=f_1(A_{i,t})),\text{do}(O_{i,t+1}=f_2(A_{i,t+1})),\text{do}(O_{i,t+2}=f_3(A_{i,t+2})))\right]$$

$$=\sum_{a_{i,t}}P(A_{i,t}=a_{i,t})\sum_{a_{i,t+1}}P(A_{i,t+1}=a_{i,t+1}|\text{do}(O_{i,t}=f_1(a_{i,t})))$$

$$\sum_{a_{i,t+2}}P(A_{i,t+2}=a_{i,t+2}|\text{do}(O_{i,t+1}=f_2(a_{i,t+1})),\text{do}(O_{i,t}=f_1(a_{i,t})))$$

$$\nabla_{f_3}P(A_{i,t+3}=j|\text{do}(O_{i,t+2}=f_3(a_{i,t+2})),\text{do}(O_{i,t+1}=f_2(a_{i,t+1})),\text{do}(O_{i,t}=f_1(a_{i,t})))$$

### G.1.3 T TIMESTEPS, GRADIENT WRT $f_k$

$$\nabla_{f_k}\mathbb{E}_{A_{i,t},...,A_{i,t+T-1}}[P(A_{i,t+T}=j|\text{do}(O_{i,t}=f_1(A_{i,t})),...,\text{do}(O_{i,t+T-1}=f_T(A_{i,t+T-1})))]$$

$$=\sum_{a_{i,t}}P(A_{i,t}=a_{i,t})\sum_{a_{i,t+1}}P(A_{i,t+1}=a_{i,t+1}|\text{do}(O_{i,t}=f_1(a_{i,t})))\cdots$$

$$\sum_{a_{i,t+k-1}}P(A_{i,t+k-1}=a_{i,t+k-1}|\text{do}(O_{i,t+k-2}=f_{k-1}(a_{i,t+k-2})),...,\text{do}(O_{i,t}=f_1(a_{i,t})))$$

$$\nabla_{f_k}\sum_{a_{i,t+k}}[P(A_{i,t+k}=a_{i,t+k}|\text{do}(O_{i,t+k-1}=f_k(a_{i,t+k-1})),...,\text{do}(O_{i,t}=f_1(a_{i,t})))$$

$$\cdots$$

$$\sum_{a_{i,t+T-1}}P(A_{i,t+T}=j|\text{do}(O_{i,t+T-1}=f_T(a_{i,t+T-1})),...,\text{do}(O_{i,t}=f_1(a_{i,t})))]$$

### G.2 STABILITY GRADIENT

### G.2.1 1 TIMESTEP

Gradient wrt $f_1$

$$\nabla_{f_1}\mathbb{E}_{\mathbf{A}_{i_2,t}}\left[d(\mathbb{P}(\mathbf{A}_{i_1,t+1}|\text{do}(\mathbf{O}_{i_2,t}=f_1(\mathbf{A}_{i_2,t}))),\mathbb{P}(\mathbf{A}_{i_1,t+1}))\right]$$

$$=\sum_{a_{i_2,t}}P(\mathbf{A}_{i_2,t}=a_{i_2,t})\nabla_{f_1}\left[d(\mathbb{P}(\mathbf{A}_{i_1,t+1}|\text{do}(\mathbf{O}_{i_2,t}=f_1(a_{i_2,t}))),\mathbb{P}(\mathbf{A}_{i_1,t+1}))\right]$$

### G.2.2 2 TIMESTEPS

- Gradient wrt $f_1$

$$\nabla_{f_1} \mathbb{E}_{\mathbf{A}_{i_2,t:t+1}} \left[ d(\mathbb{P}(\mathbf{A}_{i_1,t+2}|\mathrm{do}(\mathbf{O}_{i_2,t} = f_1(\mathbf{A}_{i_2,t})), \mathrm{do}(\mathbf{O}_{i_2,t+1} = f_2(\mathbf{A}_{i_2,t+1}))), \mathbb{P}(\mathbf{A}_{i_1,t+2})) \right]$$

$$= \sum_{a_{i_2,t}} P(\mathbf{A}_{i_2,t} = a_{i_2,t}) \nabla_{f_1} \sum_{a_{i_2,t+1}} \left[ P(\mathbf{A}_{i_2,t+1} = a_{i_2,t+1}|\mathrm{do}(\mathbf{O}_{i_2,t} = f_1(a_{i_2,t}))) \right.$$

$$\left. d(\mathbb{P}(\mathbf{A}_{i_1,t+2}|\mathrm{do}(\mathbf{O}_{i_2,t} = f_1(a_{i_2,t})), \mathrm{do}(\mathbf{O}_{i_2,t+1} = f_2(a_{i_2,t+1}))), \mathbb{P}(\mathbf{A}_{i_1,t+2})) \right]$$

- Gradient with $f_2$

$$\nabla_{f_2} \mathbb{E}_{\mathbf{A}_{i_2,t:t+1}} \left[ d(\mathbb{P}(\mathbf{A}_{i_1,t+2}|\mathrm{do}(\mathbf{O}_{i_2,t} = f_1(\mathbf{A}_{i_2,t})), \mathrm{do}(\mathbf{O}_{i_2,t+1} = f_2(\mathbf{A}_{i_2,t+1}))), \mathbb{P}(\mathbf{A}_{i_1,t+2})) \right]$$

$$= \sum_{a_{i_2,t}} P(\mathbf{A}_{i_2,t} = a_{i_2,t}) \sum_{a_{i_2,t+1}} P(\mathbf{A}_{i_2,t+1} = a_{i_2,t+1}|\mathrm{do}(\mathbf{O}_{i_2,t} = f_1(a_{i_2,t})))$$

$$\nabla_{f_2} \left[ d(\mathbb{P}(\mathbf{A}_{i_1,t+2}|\mathrm{do}(\mathbf{O}_{i_2,t} = f_1(a_{i_2,t})), \mathrm{do}(\mathbf{O}_{i_2,t+1} = f_2(a_{i_2,t+1}))), \mathbb{P}(\mathbf{A}_{i_1,t+2})) \right]$$

### G.2.3 T TIMESTEPS. GRADIENT WRT $f_k$

$$\nabla_{f_k} \mathbb{E}_{\mathbf{A}_{i_2,t:t+T-1}} \left[ d(\mathbb{P}(\mathbf{A}_{i_1,t+T}|\mathrm{do}(\mathbf{O}_{i_2,t} = f_1(\mathbf{A}_{i_2,t})), \ldots, \mathrm{do}(\mathbf{O}_{i_2,t+T-1} = f_T(\mathbf{A}_{i_2,t+T-1}))), \mathbb{P}(\mathbf{A}_{i_1,t+T})) \right]$$

$$= \sum_{a_{i_2,t}} P(\mathbf{A}_{i_2,t} = a_{i_2,t}) \sum_{a_{i_2,t+1}} P(\mathbf{A}_{i_2,t+1} = a_{i_2,t+1}|\mathrm{do}(\mathbf{O}_{i_2,t} = f_1(a_{i_2,t}))) \cdots$$

$$\sum_{a_{i_2,t+k-1}} P(\mathbf{A}_{i_2,t+k-1} = a_{i_2,t+k-1}|\mathrm{do}(\mathbf{O}_{i_2,t+k-2} = f_{k-1}(a_{i_2,t+k-2})), \ldots, \mathrm{do}(\mathbf{O}_{i_2,t} = f_1(a_{i_2,t})))$$

$$\nabla_{f_k} \sum_{a_{i_2,t+k}} \left[ P(\mathbf{A}_{i_2,t+k} = a_{i_2,t+k}|\mathrm{do}(\mathbf{O}_{i_2,t+k-1} = f_k(a_{i_2,t+k-1})), \ldots, \mathrm{do}(\mathbf{O}_{i_2,t} = f_1(a_{i_2,t}))) \right.$$

$$\left. \cdots \sum_{a_{i_2,t+T-1}} d(\mathbb{P}(\mathbf{A}_{i_1,t+T}|\mathrm{do}(\mathbf{O}_{i_2,t} = f_1(a_{i_2,t})), \ldots, \mathrm{do}(\mathbf{O}_{i_2,t+T-1} = f_T(a_{i_2,t+T-1}))), \mathbb{P}(\mathbf{A}_{i_1,t+T})) \right]$$

### G.3 REACHABILITY GRADIENT COMPUTATION FOR DETERMINISTIC (TOP-1) ITEM CHOICE

In our experiments, we set the user to always select the top item recommended by the system. While sampling items from a user's preference distribution (e.g., using Gumbel-softmax) is possible, it introduces more variability as user's now interact with a more diverse variety of item sequences, which necessitates additional parameter updates since our parameter space consists of a separate parameter for every (item, timestep) tuple. Additionally, we demonstrate below how top-1 selection affects gradient computation, with the subgradient of the item selection term reducing to 0, which is not the case for Gumbel-softmax selection. Example of 2-step gradient computation with respect to $f_1$: From G.1.1,

$$\nabla_{f_1} \mathbb{E}_{A_{i,t}, A_{i,t+1}} \left[ P(A_{i,t+2} = j|\mathrm{do}(O_{i,t} = f_1(A_{i,t})), \mathrm{do}(O_{i,t+1} = f_2(A_{i,t+1}))) \right]$$

$$= \sum_{a_{i,t}} P(A_{i,t} = a_{i,t}) \nabla_{f_1} \sum_{a_{i,t+1}} \left[ P(A_{i,t+1} = a_{i,t+1}|\mathrm{do}(O_{i,t} = f_1(a_{i,t}))) \right.$$

$$\left. P(A_{i,t+2} = j|\mathrm{do}(O_{i,t+1} = f_2(a_{i,t+1})), \mathrm{do}(O_{i,t} = f_1(a_{i,t}))) \right]$$

We focus on the terms dependent on $f_1$.

$$\nabla_{f_1} \sum_{a_{i,t+1}} \left[ P(A_{i,t+1} = a_{i,t+1}|\mathrm{do}(O_{i,t} = f_1(a_{i,t}))) \ P(A_{i,t+2} = j|\mathrm{do}(O_{i,t+1} = f_2(a_{i,t+1})), \mathrm{do}(O_{i,t} = f_1(a_{i,t}))) \right]$$

This can be written as:

$$\nabla_{f_1} \sum_{k \in [n]} (\prod_{l \neq k} \mathbb{I}_{g(k,t) \geq g(l,t)}) m(f_1, f_2)$$

Here $[n]$ denotes the list of item ids(from 1 to $n$) and $g$ is a function that takes an item and timestep as arguments and returns the user's preference score for that item at that timestep. The condition specified by the product of indicator functions is simply that $k$ is the item that the user has the highest preference score for, in other words, $k$ is the top-1 recommended item. We have also written $P(A_{i,t+2} = j | \text{do}(O_{i,t+1} = f_2(a_{i,t+1})), \text{do}(O_{i,t} = f_1(a_{i,t})))$ as a function of $f_1$ and $f_2$, $m(f_1, f_2)$. Using the product rule on this expression, we get,

$$\sum_{k \in [n]} (\prod_{l \neq k} \mathbb{I}_{g(k,t) \geq g(l,t)} \nabla_{f_1} m(f_1, f_2) + m(f_1, f_2) \nabla_{f_1} (\prod_{l \neq k} \mathbb{I}_{g(k,t) \geq g(l,t)}))$$

This reduces to:

$$\sum_{k \in [n]} (\prod_{l \neq k} \mathbb{I}_{g(k,t) \geq g(l,t)} \nabla_{f_1} m(f_1, f_2))$$

as the 0 is a subgradient of the product of indicator functions with respect to $f_1$. Therefore, in making the simplification to users only choosing the top-1 item, we do away with the gradient propagation path through the item choice itself.

