# OpenReview forum: "A Unified Causal Framework for Auditing Recommender Systems for Ethical Concerns"
_ICLR.cc/2025/Conference — Submitted to ICLR 2025_

### Official Review · Reviewer_f6Sk · 2024-10-26

**Soundness:** 2
**Presentation:** 3
**Contribution:** 2
**Rating:** 5
**Confidence:** 3

**Summary:**

In this work, the authors adopt a causal perspective on recommender system auditing and present a general method for defining auditing metrics. Within this overarching causal auditing framework, they categorize existing audit metrics. Leveraging their framework, they propose two types of metrics: future-/past-reachability and stability, which respectively measure a user's ability to influence recommendations for themselves and for other users. Additionally, they introduce a gradient-based method and a black-box method for calculating these metrics, allowing auditors to assess them at various levels of system access. Empirically, the authors demonstrate the effectiveness of their proposed metrics and use them to examine the design of recommender systems.

**Strengths:**

- Auditing recommender systems is a highly meaningful area of study, and the paper contributes valuable insights.
- The article is well-written and clearly articulated, making complex concepts accessible.
- It provides methods for auditing from both white-box and black-box perspectives, catering to different levels of system access.

**Weaknesses:**

- **W1**: **Ambiguity in Definitions**: The definitions in the article lack detailed explanations, which may lead to ambiguity. For example:
  - **Q1-1**: In Definitions 4.1 and 4.2, the authors consider only the intervention on $O_{i,t}$ without accounting for its effect on $A_{i,t+1}$. Why was this setting chosen?
  - **Q1-2**: Are Definitions 4.1 and 4.2 consistent? Specifically, does past-$k$ at time $t+k$ equal future-$k$ at time $t$? It would be helpful if the authors could address this question both intuitively and formally.

- **W2**: **Limited Analysis Scope**: The analysis in Section 5 is confined to $k=1$, representing only a special case of the broader definitions provided.
  - **Q2**: Please describe how the corresponding white-box and black-box methods would operate when $k > 1$.

- **W3**: **Practical Applicability Concerns**: There is a gap between the theoretical propositions and practical scenarios.
  - **Q3**: Proposition 5.1 requires fixing item embeddings, while Proposition 5.2 requires fixing user embeddings. Since these conditions are difficult to meet in real recommender systems, how does this gap affect practical auditing?

- **W4**: **Lack of Experimental Rationale**: Certain experimental setups lack clear justification.
  - **Q4-1**: Section 6.1 mentions different policies for future and past metrics. Why was this setup chosen? Please explain the rationale behind this decision.

- **W5**: **Incomplete Experimental Validation**:
  - **Q5-1**: The use of a single dataset limits the experimental scope and generalizability of the findings.
  - **Q5-2**: The current experiments focus on analyzing existing models within the proposed framework but do not clarify why this framework or these metrics are more valid than existing auditing methods. Additional experiments, such as straightforward case studies, are needed to further validate the framework.

**Questions:**

In addition to **Q1** to **Q5** mentioned in the Weaknesses, I have several other questions:

- **Q6**: What is the relationship between the proposed metrics and recommendation performance? Does a stronger recommendation model perform better according to these metrics?

- **Q7**: The metric comparisons in Figure 3 are described but lack corresponding explanations. For instance, why do some items show "a user’s recommended list is either heavily affected by the actions of an adversary or is minimally affected by them"?

- **Q8**: Is the time horizon parameter in the experimental parameters equivalent to $k$ in Definitions 4.1 and 4.2? If not, how is $k$ set in the experiments?

I am happy to engage in further discussion, and if these issues are addressed, I am willing to reconsider the score.

---

> ### Author Response · Authors · 2024-11-24
> **Reply to Reviewer 4**
>
> We thank the reviewer for acknowledging the importance of our work.
>
> "Auditing recommender systems is a highly meaningful area of study, and the paper contributes valuable insights."
>
> Below we address your concerns individually.
>
> **Q1-1**:
>
> **Our response**:
> In definition 4.1 (future reachability), the user is constrained to only rate the items that are recommended to them, but is allowed to assign any rating. Here, the item recommended to the user at the next time step ($A_{i,t+1}$) depends on the rating the user gives in the preceding timestep ($O_{i,t}$). Therefore, in this case, even though the intervention is only on the user action, the items the user rates in successive timesteps are dependent on these interventions.
> The decision to have the user simply rate the item that is directly recommended to them at each time step rather than having them choose the item to rate was to create a more concrete tie-in with the role of the recommender system itself in this whole process. By exclusively rating items that are recommended to them, we are able to see if the user can ‘reach’ an item simply by interacting with the recommender system. If the user was allowed to rate arbitrary items, this would mean the user can directly rate/reach the item to be reached too, which would be contrary to the problem itself.
>
> In definition 4.2 (past reachability), we fix the items the user is to rate to be the same items the user rated in the past. This again avoids the issue mentioned above and the items the user themselves chose to rate in the past are indicators of the user’s preference. This is a retrospective look at user agency, hence this metric is counterfactual. We ask the question: could the user have rated the items they already rated in a different manner and been able to reach some target item?
>
> **Q1-2**: Are Definitions 4.1 and 4.2 consistent? Specifically, does past-k at time t+k equal future-k at time t? It would be helpful if the authors could address this question both intuitively and formally."
>
> **Our response**:
> The difference between the two metrics, as detailed in the previous answer is in the choice of items the user rates in the ‘k’ timesteps before reachability is computed. In the case of future reachability, the user accepts the item recommended to them at each timestep and rates it, while in the case of past reachability, the user rates the same items they rated in the past.
>
> We elaborate on how these metrics differ and how they have their own importance under Definition 4.4 in the main paper.
>
> **Q2**: "Please describe how the corresponding white-box and black-box methods would operate when k>1."
>
> **Our Response**:
> We elaborated on this in Appendix G.1 and G.2, where we write out this expression for additional values of ‘k’.
>
> (continued in next comment)

---

> > ### Author Response · Authors · 2024-11-24
> > **Reply to Reviewer 4 (continued)**
> >
> > **Q3**: Proposition 5.1 requires fixing item embeddings, while Proposition 5.2 requires fixing user embeddings. Since these conditions are difficult to meet in real recommender systems, how does this gap affect practical auditing?"
> >
> > **Our Response**:
> > First, we clarify that the definitions of the proposed metrics don’t require specific assumptions on how the recommender is trained (i.e., not requiring the recommender system to have fixed item embeddings). Second, it is worth noting that obtaining the reachability and stability metrics involves solving a challenging optimization problem. Inherently, it is a bilevel optimization problem: the inner optimization problem is to retrain the recommender system, and the outer optimization problem is with respect to users’ own ratings (or other users’ ratings). In general, bilevel optimization is an active research area and can be quite difficult to solve [1]. To simplify this optimization problem both algorithmically and computationally, we choose to have fixed item embeddings when obtaining reachability, similar to assumptions made in [2,3,4]. If one were to solve the full bilevel optimization problem (e.g., retraining the recommender system entirely) for example when auditing the system over a longer time horizon, one may adopt black-box optimization methods similar to the one discussed in Sec 5.2.
> >
> > [1 ]B. Colson, et al. "An overview of bilevel optimization." In Annals of operations research (2007).
> > [2] Sirui Yao, Yoni Halpern, Nithum Thain, Xuezhi Wang, Kang Lee, Flavien Prost, Ed H. Chi, Jilin Chen, and Alex Beutel. Measuring recommender system effects with simulated users. CoRR, abs/2101.04526, 2021. URL https://arxiv.org/abs/2101.04526.
> > [3] Mihaela Curmei, Sarah Dean, and Benjamin Recht. Quantifying availability and discovery in recommender systems via stochastic reachability. In Proceedings of the 38th International Conference on Machine Learning, pp. 2265–2275. PMLR, 2021.
> > [4] Sarah Dean, Sarah Rich, and Benjamin Recht. Recommendations and user agency: The reachability of collaboratively-filtered information. In Proceedings of the 2020 Conference on Fairness, Accountability, and Transparency, FAT* ’20, pp. 436–445, New York, NY, USA, 2020. Association for Computing Machinery.
> >
> > **Q4-1**: "Section 6.1 mentions different policies for future and past metrics. Why was this setup chosen? Please explain the rationale behind this decision."
> >
> > **Our Response**:
> > We justify our choice for a different selection rule for future facing metrics in Appendix C. Future-k metrics are more computationally intensive as they require optimizing k×|V| parameters to account for all possible item trajectories. However, we made this more tractable by using a deterministic (top-1) recommendation policy rather than stochastic sampling. We detail the effects in Appendix G.3.
> >
> > **Q5-1**: "The use of a single dataset limits the experimental scope and generalizability of the findings."
> >
> > **Our Response**:
> > We agree that additional experiments would provide for a valuable comparison. However, we would like to clarify that one of our key contributions is providing a framework for quantitatively assessing user agency metrics. The framework applies to a broad class of recommender systems and is domain agnostic. The framework requires only three fundamental components to define a causal metric: (1) an intervention, (2) an outcome, and (3) a functional that maps the outcome to a real value. These components are abstract and can be instantiated in any application domain.  While we demonstrate the framework's utility through movie recommendations, the mathematical formulation itself makes no domain-specific assumptions.
> >
> >
> > (continued in next comment)

---

> > > ### Author Response · Authors · 2024-11-24
> > > **Reply to Reviewer 4 (continued)**
> > >
> > > **Q5-2**: "The current experiments focus on analyzing existing models within the proposed framework but do not clarify why this framework or these metrics are more valid than existing auditing methods. Additional experiments, such as straightforward case studies, are needed to further validate the framework."
> > >
> > > **Our Response**:
> > > We note the general deficiencies we see with existing auditing metrics in Section 3.2, primarily that they are association-based. Specifically, we cite metrics like diversity and popularity that are defined over an observational distribution.
> > > In Section 4, we elaborate on how the notion of user agency ties into ethical concerns associated with recommender systems: primarily the formation of filter bubbles that can serve to only reinforce biases and insulate users from opposing viewpoints, and the possibility of adversaries having an outsized ability to manipulate the recommender system. In Section 2, we mention how studying user agency requires answering causal questions pertaining to how a recommender would respond to interventions made in the user’s behavior over time, which cannot be answered by association-based metrics alone.
> > > We also talk about the deficiencies of existing interventional metrics in Section 2: they are often limited in the kind of interventions they consider, with few works tapping into new types of interventions and ethical concerns. In addition, most are one-step metrics, ignoring the recommender-user interaction dynamics over time.
> > > Overall, to our knowledge, this is the first work that concretely proposes a causal framework for auditing recommender systems, and allows for a general procedure for defining auditing metrics (Section 3.2) besides the ones we define in Section 4.
> > >
> > > **Q6**: "What is the relationship between the proposed metrics and recommendation performance? Does a stronger recommendation model perform better according to these metrics?"
> > >
> > > **Our Response**:
> > > We argue that the proposed metrics act as a separate measure to evaluate the ‘strength’ of a recommender system. A recommender system that performs extremely well on traditional accuracy/recall-style evaluation metrics may not perform well on the auditing metrics we define. For instance, these traditional evaluation metrics cannot directly detect whether or not a user is in a filter bubble since this would require an interventional perspective. A recommender system with such an issue can still perform highly on these metrics as the user is sucked deeper into the bubble and is unaware that the bubble even exists since they are insulated from other items.
> > > A strong recommender system should perform highly on these traditional metrics in addition to the new metrics we propose, which value the capacity of the recommender system to promote user agency.
> > >
> > > **Q7**: "The metric comparisons in Figure 3 are described but lack corresponding explanations. For instance, why do some items show 'a user’s recommended list is either heavily affected by the actions of an adversary or is minimally affected by them'?"
> > >
> > > **Our Response**:
> > > Figure 3 has 2 sets of plots. The reachability plots are scatterplots where each dot represents a (user, item) pair. Some (user, item) pairs have higher base reachability (x-coordinate) than others, just by virtue of the initial match between the user and item as perceived by the recommender system. Each (user, item) pair then also has a different max-reachability (y-coordinate), because of the differences in final match between the user and item as perceived by the recommender system after the primal perturbation.
> > >
> > > Similarly, the stability plots are histograms of stability values for each (user, adversary) pair. Some adversaries have a greater effect on the recommendations of some target users than other adversaries; for instance, if a user very similar to the target user changes their ratings, the recommendations of the target user are more likely to change than if someone whose preferences have little correlation with the preferences of the target user changes their ratings. We also see that this distribution is mostly bimodal, implying that for these recommender systems on this dataset, for each user, every other user can broadly either be an effective adversary or an ineffective adversary.
> > >
> > > **Q8**: "Is the time horizon parameter in the experimental parameters equivalent to k in Definitions 4.1 and 4.2? If not, how is k set in the experiments?"
> > >
> > > **Our Response**:
> > > Yes, the time-horizon parameter refers to ‘k’ in Definitions 4.1 and 4.2.

---

> > > > ### Comment · Reviewer_f6Sk · 2024-11-25
> > > >
> > > > Thanks for the authors' response. While some of my concerns have been addressed, there are still unresolved issues that need further clarification:
> > > >
> > > > ### Response to **A1-1**
> > > > You seem to have misunderstood **Q1-1**. As you mentioned, "the item recommended to the user at the next time step ($A_{i,t+1}$) depends on the rating the user gives in the preceding timestep ($O_{i,t}$)," but in your Equation 1, you only consider:
> > > > $
> > > > \mathbb{P}(A_{i,t+k} = j \mid do(O_{i,t+k-1} = f(A_{i,t+k-1})), do(O_{i,t+k-2} = f(A_{i,t+k-2})), \cdots)
> > > > $
> > > > This does not account for the effect of $do(O_{i,t+k-2} = f(A_{i,t+k-2}))$ on $A_{i,t+k-1}$. That is, after the intervention $O_{i,t+k-2}$, $A_{i,t+k-1}$ remains in the state of an un-changed recommendation. My question in **Q1-1** was: why did you not consider the impact of $do(O_{i,t+k-2} = f(A_{i,t+k-2}))$ on $A_{i,t+k-1}$?
> > > >
> > > > ### Response to **A1-2**
> > > > Based on **A1-2**, can I understand that **future reachability** considers the effect of $do(O_{i,t+k-2} = f(A_{i,t+k-2}))$ on $A_{i,t+k-1}$, whereas **past reachability** does not? If so:
> > > > 1. Why is this distinction not reflected in Equation 1?
> > > > 2. What is the practical significance of **past reachability**? Why analyze the expected reachability of an item under unchanging recommendations? What does this imply in real-world scenarios?
> > > >
> > > > ### Response to **A2**
> > > > My question is: when $k>1$, do Equation 5 and the formulas in Section 5.2 need to be adjusted? Or would this require retraining to evaluate the metrics? I would like the authors to explain this in detail.
> > > >
> > > > ### Response to **A3**
> > > > I understand that you made assumptions to facilitate analysis. However, I want to know how significant the impact of these assumptions would be on practical auditing. Previous works using similar assumptions do not necessarily imply that these assumptions will not affect the auditing process.
> > > >
> > > > ### Response to **A5-1**
> > > > I understand that your method is not limited to specific datasets. However, my concern is that some of the conclusions you draw in the experimental section, such as “As we increase $\beta$, or decrease the stochasticity of the system, user recommendations tend to become more stable,” are data-dependent. If consistent results cannot be achieved across multiple datasets, these conclusions will be hard to accept. As other reviewers have also raised this point, I strongly encourage you to include results from additional datasets.
> > > >
> > > > ---
> > > >
> > > > Considering the authors’ response, I am willing to raise my score for this paper. However, as the above issues, particularly **Q1-1**, **Q1-2**, and **Q5-1**, have not been adequately addressed, I am unable to provide a positive score for this submission.

---

> > > > > ### Author Response · Authors · 2024-12-01
> > > > > **Reply to Reviewer f6Sk**
> > > > >
> > > > > **Response to “Response to A1-1”**:
> > > > > Your interpretation  written in A1-2 is correct. For interventional metrics (the future-facing metrics), the item rated by the user in the next time step is directly the item recommended to the user. The recommendation in subsequent timesteps depends on the rating given in the previous timestep and therefore considers the impact of the do-intervention that you have mentioned. On the other hand, for counterfactual metrics (the past-facing metrics), the items the user rates are fixed to be the actual items they rated in the past (the user’s true history), and aren’t affected by the user’s intervention in the previous timestep. Note that the recommender system is still updated after every interaction.
> > > > >
> > > > > **Response to “Response to A1-2”**:
> > > > > Your understanding is correct. Comparing Equation 1 and Equation 2, the difference is that in Equation 1, the user’s choice follows the recommendation dynamics, which allows A_{i, t+k-1} to be influenced by previous outcomes. The recommender system is updated (through updating the user vector) after every interaction the user has, and so the item the user is recommended at time t+k-1 depends on the rating the user gives to the item recommended to them at time t+k-2 (this is the do-intervention that you mention). Meanwhile, in Equation 2, the recommendation history is set to be the user’s factual history and the users are only allowed to change their feedback for them. We specify this in the manuscript where after Definition 4.1, we point out that the intervention $f_{t'}(A_{i,t + t'- 1})$ would affect the distribution of $A_{i,t + t'}$, the recommendations in the next time step, in the case of future reachability.
> > > > > The practical significance of past reachability is that it helps answer the crucial question: "Could a user have received different recommendations in the present if they had rated their historically viewed items differently?"  Under Definition 4.4, we provide a brief explanation of the differences between past and future facing metrics, and we reiterate the utility of past-facing metrics mentioned there here: “Past-/counterfactual metrics focus on how user behavior (e.g., the items a user chose to rate) contributes to the recommendations they receive in the present. For example, consider a social media user who primarily receives recommendations for cat videos in the present. Counterfactual/past metrics help us understand whether the narrowness in the recommendations can be attributed to the recommendation system, or the user’s behavior in the past, which imply vastly different conclusions in terms of user agency. If engaging with cat videos unfavorably in the past would have led to more diverse recommendations in the present, the observed narrow recommendations do not imply a violation of user agency.”
> > > > > We noted in the earlier reply how it is necessary to fix an item selection rule (otherwise, the user could just select the item they have to reach), and setting these items to be the actual sequence of items the user rated is a natural choice since they reflect the preferences of the user to a large extent. An intuitive distinction with future-facing metrics is that for past-facing metrics, the user’s choice of item essentially reflects their true preference (since it was an item they actually selected), while for future-facing metrics, the user’s choice of item reflects their preference as perceived by the recommender system. Both points of view offer us distinct, but related, insights into user agency.

---

> > > > > > ### Author Response · Authors · 2024-12-01
> > > > > > **Reply to Reviewer f6Sk**
> > > > > >
> > > > > > **Response to "Response to A2"**:
> > > > > >
> > > > > > The expressions in G.1 and G.2 are the corresponding equations to the first equation in Section 5 (before Section 5.1) for 'k' taking values other than 1. These equations essentially tell us the gradient terms we need to compute (or need access to).
> > > > > >
> > > > > >
> > > > > > For k=1, this term is:
> > > > > >
> > > > > >
> > > > > > $\nabla_{f_1} \mathbb{P}(A_{i, t+1} = j | \text{do}(O_{i,t} = f_1(A_{i,t})))$
> > > > > >
> > > > > >
> > > > > > While for k=2 (for example), these terms, as shown in Section G.1, are:
> > > > > >
> > > > > >
> > > > > > $\nabla_{f_1}  \sum_{a_{i,t+1}} \left[ P(A_{i,t+1} = a_{i,t+1} | \text{do}(O_{i,t} = f_1(a_{i,t}))) \cdot P(A_{i,t+2} = j | \text{do}(O_{i,t+1} = f_2(a_{i,t+1})), \text{do}(O_{i,t} = f_1(a_{i,t}))) \right]$
> > > > > >
> > > > > >
> > > > > > and
> > > > > >
> > > > > >
> > > > > > $\nabla_{f_2} P(A_{i,t+2} = j | \text{do}(O_{i,t+1} = f_2(a_{i,t+1})), \text{do}(O_{i,t} = f_1(a_{i,t})))$
> > > > > >
> > > > > >
> > > > > > after simplifying
> > > > > >
> > > > > >
> > > > > > $\nabla_{f_1} E_{A_{i,t}, A_{i,t+1}} \left[ P(A_{i,t+2} = j | \text{do}(O_{i,t} = f_1(A_{i,t})), \text{do}(O_{i,t+1} = f_2(A_{i,t+1}))) \right]$
> > > > > >
> > > > > >
> > > > > > and
> > > > > >
> > > > > >
> > > > > > $\nabla_{f_2} E_{A_{i,t}, A_{i,t+1}} \left[ P(A_{i,t+2} = j | \text{do}(O_{i,t} = f_1(A_{i,t})), \text{do}(O_{i,t+1} = f_2(A_{i,t+1}))) \right]$ respectively.
> > > > > >
> > > > > >
> > > > > > Equation 5 simply gives us the probability of item 'j' being recommended to user 'i'. In the stochastic setting we consider, this is proportional to $\exp(\beta \mathbf{p}_i^\top \mathbf{q}_j)$, where $p_i$ is the user vector corresponding to 'i' after the specified perturbation has been made, and $q_j$ is the item vector corresponding to 'j'.
> > > > > >
> > > > > >
> > > > > > Equation 5 hides away all the complexity within $p_i$ and $q_j$. For reachability, the user vector $p_i$ is updated after every interaction. We specify this under equation 5 where we note that these user and item vectors are the updated user and item vectors depending on the intervention. For k=1, this update is after only one interaction. For k=n, this update is after 'n' interactions. So the relation in Equation 5 is valid for all k, only $p_i$ or $q_j$ will change depending on the specified do-condition.
> > > > > >
> > > > > >
> > > > > > So this would take the form:
> > > > > >
> > > > > >
> > > > > > $\mathbb{P}(A_{i, t+k} = j | \text{do}(O_{i,t} = f_1(A_{i,t})), \ldots, \text{do}(O_{i,t+k-1} = f_k(A_{i,t+k-1}))) \propto \exp(\beta \mathbf{p}_i^\top \mathbf{q}_j)$
> > > > > >
> > > > > >
> > > > > > Where $p_i$ is now the updated user vector after simulation according to the specified do-conditions.
> > > > > >
> > > > > >
> > > > > > Similarly, in Section 5.2, for other values of 'k', the do-condition will be different as specified in Definition 4.1:
> > > > > >
> > > > > >
> > > > > > $\mathbb{P}(A_{i, t+k} = j | \text{do}(O_{i,t} = f_1(A_{i,t})), \ldots, \text{do}(O_{i,t+k-1} = f_k(A_{i,t+k-1})))$
> > > > > >
> > > > > >
> > > > > > but this probability will still be given by Equation 5, with the user vector $p_i$ correspondingly updated after running the simulation according to the specified do-condition.
> > > > > >
> > > > > >
> > > > > > **Response to "Response to A3"**:
> > > > > >
> > > > > >
> > > > > > As you point out, these assumptions facilitate the analysis, providing clean theoretical results to obtain the metrics. For practical auditing, with no assumption on how the model is updated, our proposed methodology for obtaining these metrics still works (i.e., solving the optimization problems in equation 1,2,3,4. Section 5 shows how this looks for equation 1 with k=1, and the specific gradient term we need to compute. Similarly, Section 5.2 shows how this is done for black-box access). We only use the assumptions for the experiments in the paper because of relatively simpler implementations.
> > > > > > For practical auditing, with the current approach of only updating one vector, we only foresee a potential negative impact for recommender systems that are retrained after every few interactions, and even then, only if successive models differ significantly from each other.
> > > > > > Since practical recommender systems are not fully retrained after every single interaction, and only after fixed intervals, the auditing procedure is still valid between these intervals and accounts for small changes brought about by every interaction without the need for full retraining, albeit with some noise.
> > > > > >
> > > > > > We are actively working to conduct the additional experiments mentioned in **"Response to A5-1"** and aim to include the results in the final version.

---

### Official Review · Reviewer_C1ia · 2024-10-31

**Soundness:** 2
**Presentation:** 2
**Contribution:** 2
**Rating:** 5
**Confidence:** 5

**Summary:**

In this work, authors pay attention to recommender system auditing from a causal perspective, and point out the lack of metrics for auditing user agency for the recommendation process. Therefore, two metrics are proposed, including future- and past-reachability and stability, which can measure the impact of users on their own and other users. To calculate these metrics, the authors also design a gradient-based and a black box approach.

**Strengths:**

S1- This paper provides comprehensive details on the background of the problem.

S2- The authors give detailed experiment settings which improves the reproducibility.

**Weaknesses:**

W1-The motivation of this paper is not quite clear. For example, what’s the actual relationship between user agency and ethical concerns?

W2-The experiments are only conducted on ML-1M, which are insufficient to explain the universality of the conclusions since the recommendation senarios are diverse. Experiments on at leasr one dataset from other recommendation senarios are needed.

W3- In Figure 3 for the distribution of past instability values, for MF, Past-5 shows lower proportion of 0.0 than Past-1, but for RRN, Past-5 presents higher proportion of 0.0 than Past-1. Could you please explain the reason for this contrary result?

**Questions:**

Please see them in the weaknesses.

---

> ### Author Response · Authors · 2024-11-24
> **Reply to Reviewer 3**
>
> We thank you for your feedback. Below we address your concerns individually.
>
> **Weakness 1**: "The motivation of this paper is not quite clear. For example, what’s the actual relationship between user agency and ethical concerns?"
>
> **Our Response**:
> We elaborated on how user agency connects to ethical concerns for recommender systems in Section 4 (L195 - L207). While, to the best of our knowledge, little work on quantifying user autonomy and agency has been conducted, there is a rich line of qualitative work emphasizing the importance of user agency in recommender systems. These works conceptualize user agency as a user’s power over their own recommendations vs recommendations being driven by external forces like other users' behaviors or algorithmic profiling [2]. User agency can be compromised in various ways, several of which we target with the metrics we propose:
>
> -First, recommender systems can enforce filter bubbles that restrict users from diverse content feeds and amplify biases [1]. Looking at this problem from the context of the reachability metric we propose, we consider scenarios where certain groups of items are unreachable for a user in spite of them taking actions specifically in order to attempt to reach these items. This implies the existence of a filter bubble, which could be harmful. Without counterfactual and interventional metrics as the ones we proposed, it is hard to attribute the effect of the algorithm versus the effect of user's own behaviors on creating filter bubbles, which motivates our causal definitions of reachability.
>
> -Second, recommendation algorithms can be vulnerable to strategic behaviors and adversarial attacks that alter recommendations for unrelated users [1]. For example, consider content duplication attacks on e-commerce platforms [3]. In this setting, providers game the recommendation system by duplicating item listings with little to no changes to maximize the probability of recommendation. Maintaining user agency over their own recommendations in this scenario requires stability in the recommendations (e.g., recommendations of users cannot be manipulated easily), motivating our definitions of counterfactual and interventional stability metrics. As such, user agency ties directly to the core ethical concerns of recommender systems.
> [1]Silvia Milano, Mariarosaria Taddeo, and Luciano Floridi. Recommender systems and their 429 ethical challenges. AI amp; SOCIETY, 35(4):957–967, February 2020.
> [2]K. De Vries. Identity, profiling algorithms and a world of ambient intelligence. In Ethics and information technology (2010).
> [3]M. Frobe, et al. The effect of content-equivalent near-duplicates on the evaluation of search engines. In European Conference on Information Retrieval (2020).
>
> **Weakness 2**: "The experiments are only conducted on ML-1M, which are insufficient to explain the universality of the conclusions since the recommendation scenarios are diverse. Experiments on at least one dataset from other recommendation scenarios are needed."
>
> **Our response**:
> We would like to clarify that our framework's theoretical foundations are domain-agnostic by design. The framework requires only three fundamental components to define a causal metric: (1) an intervention, (2) an outcome, and (3) a functional that maps the outcome to a real value. These components are abstract and can be instantiated in any application domain. One of our key contributions is providing a framework for quantitatively assessing user agency metrics. The framework applies to a broad class of recommender systems and is domain agnostic. While we demonstrate the framework's utility through movie recommendations, the mathematical formulation itself makes no domain-specific assumptions.
>
> **Weakness 3**: "In Figure 3 for the distribution of past instability values, for MF, Past-5 shows lower proportion of 0.0 than Past-1, but for RRN, Past-5 presents higher proportion of 0.0 than Past-1. Could you please explain the reason for this contrary result?"
>
> **Our response**:
> The main conclusion we draw regarding instability is that allowing malicious actors (adversaries) to exist on the platform for longer periods leads to more harm as they are allowed more opportunities to manipulate the recommender system. This observation is echoed in both plots: for the RRN case, mean instability values for Past-5 Instability are still larger than those for Past-1 Instability because of additional concentration away from 0, which is a lot more non-existent for Past-2 Instability.

---

> > ### Author Response · Authors · 2024-12-04
> > **Follow-up comment by Authors**
> >
> > Dear Reviewer C1ia, we hope our rebuttal has addressed your concerns. We value your feedback and would appreciate any further questions or thoughts before the end of the rebuttal period.

---

### Official Review · Reviewer_FqFb · 2024-11-04

**Soundness:** 3
**Presentation:** 2
**Contribution:** 3
**Rating:** 3
**Confidence:** 3

**Summary:**

The paper proposes a unified causal framework for auditing recommender systems, specifically to address ethical concerns such as user agency, stability, and reachability. It categorizes auditing metrics from a causal perspective and introduces two key metrics, past- and future-reachability, and stability, which measure a user’s ability to influence recommendations. The empirical studies evaluate the metrics on different recommender models, highlighting the trade-offs between user influence on recommendations and system stability.

**Strengths:**

The causal approach offers a novel way to address ethical issues, providing a structured method for defining and calculating user-centric metrics.

Offering both gradient-based and black-box methods for metric computation enables broader application

**Weaknesses:**

The framework’s reliance on specific causal assumptions and models,this may reduce its generalizability across diverse recommender systems.

The paper lacks a discussion about the differences between recommendation systems.

**Questions:**

What are the differences and impacts of applying this model to various recommendation models?

---

> ### Author Response · Authors · 2024-11-24
> **Reply to Reviewer 2**
>
> We thank you for acknowledging the novelty of our work:
>
> "The causal approach offers a novel way to address ethical issues, providing a structured method for defining and calculating user-centric metrics."
>
> Below we address your concerns.
>
> **Weakness 1**: The framework’s reliance on specific causal assumptions and models, this may reduce its generalizability across diverse recommender systems.
>
> **Our Response**:
> Making causal assumptions is necessary since the metrics we define are causal and identifying/estimating causal effects requires us to make certain assumptions. We want to emphasize that the framework we propose **does not** rely on specific assumptions about the recommender systems it is used to audit. The causal graph in Figure 1 is general for all recommender systems as all recommender systems involve providing recommendations, collecting (possibly empty) user feedback, and then using the user-recommendation history to determine future recommendations. Our causal graph captures this, laying out the dependency among the factors involved in this recommendation process, **without** assuming anything about the recommendation algorithm itself.
>
> The sequential dependency of these factors can be viewed as causal assumptions. However, the dependency of these factors can only be the presented way in Figure 1 as in reality, these factors show up in a sequential order. That is, a user can only provide feedback on a particular recommendation after a recommendation is given to them,
> and the next-stage recommendation can only depend on past (possibly empty) user feedback instead of future ones.
>
> **Weakness 2**: "The paper lacks a discussion about the differences between recommendation systems."
> **Question 1**: "What are the differences and impacts of applying this model to various recommendation models?"
>
> **Our Response**:
> As we noted above, our general causal auditing framework apply to all recommender systems. To illustrate the usage of our proposed metrics, we evaluated two types of recommender systems empirically: a Matrix Factorization based recommender and a Recurrent Recommender Network. In our analysis, we find that the Matrix Factorization based recommender promotes greater user-item reachability but has less user-adversary stability as compared to its RNN based counterpart. While the operationalization details may vary slightly between different recommender systems (as detailed in Section 5), the framework itself (Section 3) and the metrics we define (Section 4) are model-agnostic.

---

> > ### Comment · Reviewer_FqFb · 2024-12-03
> >
> > Thank you for the rebuttal. I would like to keep my current score.

---

### Official Review · Reviewer_WsZQ · 2024-11-06

**Soundness:** 3
**Presentation:** 3
**Contribution:** 2
**Rating:** 5
**Confidence:** 3

**Summary:**

The paper presents a unified causal framework for auditing recommender systems with focus on user agency. The authors make three main contributions:
1. A general causal framework that formalizes interventional and counterfactual metrics for auditing recommender systems.
2. Two novel classes of metrics - reachability and stability, to measure user agency while accounting for recommendation dynamics.
3. Efficient computational methods for these metrics under different levels of access to the recommender system.

The framework is evaluated empirically using both matrix factorization and recurrent neural network based recommenders, showcasing interesting trade-offs between stability and reachability.

**Strengths:**

1. The technical claims and methodology are very well-supported. The causal framework is rigorously developed with clear mathematical formulations. The empirical evaluation is comprehensive, with well-designed ablation studies showing impact of various stochasticity levels, time horizon lengths and model architecture choices.

2. Novel formalization of reachability and stability metrics presented capture both immediate and long-term effects, handle multi-step recommendation dynamics and account for both user and adversary perspectives.

3. The paper is generally well-written and logically structured. The causal framework is presented clearly with helpful examples.

**Weaknesses:**

1. The assumption of static user/item embeddings during gradient computation could be better justified. Additional experiments showing impact of this simplification would be valuable.

2. The empirical evaluation focuses on movie recommendations - testing on other domains (e.g. social media, e-commerce, etc.) would strengthen the framework's generalizability claims.

3. The choice of distance metrics for stability measures (L2 distance) could be better justified. Adding discussion of metric sensitivity to adversarial perturbations and analysis of the relationship between local and global notions of reachability would be useful.

4. The paper presents limited discussion of computational complexity and scalability analysis, particularly for large-scale recommender systems. The paper could analyze how the methods scale with number of users, items and time horizon.

**Questions:**

1. How does the computational complexity scale with the number of users, items and time horizon? What are recommended approaches for large-scale recommender systems?

2. What are the practical implications of assuming static embeddings during gradient computation? How would the results change with full retraining?

3. Could the framework be extended to handle more complex recommendation scenarios like slate recommendations or contextual bandits?

---

> ### Author Response · Authors · 2024-11-24
> **Reply to Reviewer 1**
>
> We thank you for appreciating the rigor of our work:
>
> "The technical claims and methodology are very well-supported. The causal framework is rigorously developed with clear mathematical formulations. The empirical evaluation is comprehensive, with well-designed ablation studies showing impact of various stochasticity levels, time horizon lengths and model architecture choices."
>
> We address your concerns individually as follows.
>
> **Weakness 1**: "The assumption of static user/item embeddings during gradient computation could be better justified. Additional experiments showing impact of this simplification would be valuable."
> **Question 2**: "What are the practical implications of assuming static embeddings during gradient computation? How would the results change with full retraining?"
>
> **Our Response**:
> First, we clarify that the definitions of the proposed metrics don’t require specific assumptions on how the recommender is trained (i.e., not requiring the recommender system to have fixed item embeddings). Second, it is worth noting that obtaining the reachability and stability metrics involves solving a challenging optimization problem. Inherently, it is a bilevel optimization problem: the inner optimization problem is to retrain the recommender system, and the outer optimization problem is with respect to users’ own ratings (or other users’ ratings). In general, bilevel optimization is an active research area and can be quite difficult to solve [1]. To simplify this optimization problem both algorithmically and computationally, we choose to have fixed item embeddings when obtaining reachability, similar to assumptions made in [2,3,4]. If one were to solve the full bilevel optimization problem (e.g., retraining the recommender system entirely) for example when auditing the system over a longer time horizon, one may adopt black-box optimization methods similar to the one discussed in Sec 5.2.
>
> [1 ]B. Colson, et al. "An overview of bilevel optimization." In Annals of operations research (2007).
> [2] Sirui Yao, Yoni Halpern, Nithum Thain, Xuezhi Wang, Kang Lee, Flavien Prost, Ed H. Chi, Jilin Chen, and Alex Beutel. Measuring recommender system effects with simulated users. CoRR, abs/2101.04526, 2021. URL https://arxiv.org/abs/2101.04526.
> [3] Mihaela Curmei, Sarah Dean, and Benjamin Recht. Quantifying availability and discovery in recommender systems via stochastic reachability. In Proceedings of the 38th International Conference on Machine Learning, pp. 2265–2275. PMLR, 2021.
> [4] Sarah Dean, Sarah Rich, and Benjamin Recht. Recommendations and user agency: The reachability of collaboratively-filtered information. In Proceedings of the 2020 Conference on Fairness, Accountability, and Transparency, FAT* ’20, pp. 436–445, New York, NY, USA, 2020. Association for Computing Machinery.
>
> **Weakness 2**: "The empirical evaluation focuses on movie recommendations - testing on other domains (e.g. social media, e-commerce, etc.) would strengthen the framework's generalizability claims."
>
> **Our Response**:
> We agree with your suggestion that additional evaluation could strengthen the framework’s generalizability claims. However, we would like to clarify that our framework's theoretical foundations are domain-agnostic by design. The framework requires only three fundamental components to define a causal metric: (1) an intervention, (2) an outcome, and (3) a functional that maps the outcome to a real value. These components are abstract and can be instantiated in any application domain. Our contribution is a framework for defining new metrics - One of our key contributions is providing a framework for quantitatively assessing user agency metrics. The framework applies to a broad class of recommender systems and is domain agnostic. While we demonstrate the framework's utility through movie recommendations, the mathematical formulation itself makes no domain-specific assumptions.
>
> (response continued in next comment)

---

> ### Author Response · Authors · 2024-11-24
> **Reply to Reviewer 1 (continued)**
>
> **Weakness 3**: "The choice of distance metrics for stability measures (L2 distance) could be better justified. Adding discussion of metric sensitivity to adversarial perturbations and analysis of the relationship between local and global notions of reachability would be useful."
>
> **Our Response**:
> We have used Hellinger Distance as the distance metric for stability plots. We note that the definition of stability allows for any metric that measures the distance between two probability distributions. In Appendix C, we specifically show how using L2 distance, and other metrics sharing some similar properties, can make computing past-stability a quasi-convex problem where the solution can be easily obtained by simply checking boundary points of the domain, justifying its use for the paper.
>
> Both reachability and stability are already defined as an optimization problem that finds the optimal perturbation for a specific objective. For reachability, we solve for the best perturbation a user can make to maximize their probability of reaching a desired item. For stability, we find the optimal perturbation an adversary can make to maximize the distance between a user's initial and final recommendation distributions. During the audit, this is the only perturbation that takes place - the recommender system then evolves according to its natural dynamics. Therefore, analyzing sensitivity to additional adversarial perturbations is redundant, as we've already characterized the maximum impact possible within our well-defined perturbation space.
>
> Regarding the relationship between local and global reachability - our framework actually captures both through different parameter settings. The past-k and future-k reachability metrics can examine both local effects (when k=1 or with small rating changes) and global effects (with larger k values or more substantial rating modifications). This flexibility allows auditors to examine both incremental changes and more dramatic shifts in recommendation patterns.
>
> **Weakness 4**: "The paper presents limited discussion of computational complexity and scalability analysis, particularly for large-scale recommender systems. The paper could analyze how the methods scale with number of users, items and time horizon."
> **Question 1**: "How does the computational complexity scale with the number of users, items and time horizon? What are recommended approaches for large-scale recommender systems?"
>
> **Our Response**:
> As mentioned, our methods scale with three key factors: number of users (n), items (m), and time horizon (k). For both past-k metrics, we optimize k parameters (one per timestep). Future-k metrics are more computationally intensive as they require optimizing k×|V| parameters to account for all possible item trajectories. However, we made this more tractable by using a deterministic (top-1) recommendation policy rather than stochastic sampling. We detail the effects in Appendix G.3. In practice, the metrics can be computed on sampled subsets of users and items for large-scale systems while still providing meaningful audit insights.
>
> **Question 3**: "Could the framework be extended to handle more complex recommendation scenarios like slate recommendations or contextual bandits?"
>
> **Our Response**:
> Yes, the framework can be extended to handle the more complex recommendation scenarios mentioned in the question. The general framework itself (Section 3) does not depend on specifics on the recommendation algorithms. Operationalizing for computing the metrics is the part that relies on these specifics. For instance, in slate recommendation, now instead of computing the probability of the item under consideration being the recommended item (for reachability), we will be optimizing the probability of the item being part of the slate of recommended items instead.

---

> > ### Author Response · Authors · 2024-12-04
> > **Follow-up comment by Authors**
> >
> > Dear Reviewer WsZQ, we hope our rebuttal has addressed your concerns. We value your feedback and would appreciate any further questions or thoughts before the end of the rebuttal period.

---

### Author Response · Authors · 2024-11-24
**Summary of Primary Conceptual Contributions of our work**

We are very thankful for the reviewers' feedback and wanted to outline the key conceptual contributions of our paper to clarify common questions raised by some reviewers.

We note that while much attention has been given to auditing various ethical concerns of recommender systems, there has been comparatively little work conducted on measuring user agency. This gap is particularly notable given the rich line of qualitative work emphasizing the importance of user agency [1]. User agency is a user’s power over their own recommendations compared to recommendations being driven by external forces like other users’ behaviors or algorithmic profiling [2]. It can be compromised in various ways, several of which we target with the metrics we propose.
Two primary ethical concerns associated with the violation of user agency that we study here are the existence of filter bubbles and susceptibility to adversarial attacks. Both pose important ethical threats, as filter bubbles reinforce biases and restrict opposing viewpoints while adversarial attacks essentially game the recommender system to fulfill an objective.

We illustrate how existing metrics used to audit recommender systems are either 'association-based' and cannot audit user agency, or make assumptions that are not in line with the true dynamics of the recommender system. The unified causal framework that we propose and the two classes of metrics we introduce resolve these issues.

In summary,

1) We provide a general causal framework for defining new causal metrics and categorizing existing
metrics for auditing recommender systems in a principled manner (Section 3)

2) Using our proposed framework, we develop two classes of metrics for measuring user agency while
accounting for the dynamics of the recommendation process: past- and future-reachability and
stability (Section 4). We provide effective ways to compute the metrics, allowing the auditor to
have different levels of access to the systems (Section 5).

3) Empirically, we investigate two common classes of recommender systems in terms of our proposed
user agency metrics and found that higher stochasticity in a recommender system will help with
stability but harm reachability (Section 6).

[1] Silvia Milano, Mariarosaria Taddeo, and Luciano Floridi. Recommender systems and their ethical
challenges. AI & SOCIETY, 35(4):957–967, Feb 2020a.

[2] Katja de Vries. Identity, profiling algorithms and a world of ambient intelligence. Ethics and
Information Technology, 12(1):71–85, January 2010.

---

### Meta-Review · Area_Chair_oYS8 · 2024-12-19

**Metareview:**

This paper presents a causal framework for auditing recommender systems with a focus on ethical considerations such as user agency, stability, and reachability. It introduces two novel classes of metrics: future- and past-reachability and stability, which measure a user’s influence over their own recommendations and others’. The authors also propose two computational approaches, gradient-based and black-box, for calculating these metrics. Empirical evaluations demonstrate the utility of these metrics using matrix factorization and recurrent recommender network models, revealing insights into the trade-offs between stability and reachability.

The paper has several strengths. It tackles an important and underexplored problem in auditing recommender systems from a causal perspective, offering a framework that has the potential to generate new insights into user-centric evaluation metrics. The proposed metrics are novel, and the framework itself is methodologically rigorous and well-motivated. Additionally, the authors provide detailed explanations and experiments, improving the paper’s clarity and reproducibility.

However, the paper has notable weaknesses that undermine its potential contribution. First, the motivation for focusing on user agency as an ethical concern, while mentioned, lacks a compelling connection to practical scenarios and actionable insights. The authors do not sufficiently justify the real-world impact of their metrics, especially when tied to the ethical concerns they aim to address. Second, the experimental evaluation is narrow in scope, relying solely on the ML-1M dataset, which limits the generalizability of the findings across diverse recommendation domains. Reviewers consistently raised concerns about the lack of validation on other datasets, as this significantly impacts the credibility of the conclusions. Additionally, the framework relies on several assumptions, such as static user and item embeddings, that simplify the computational challenges but reduce the practical applicability of the methods for real-world, large-scale systems. This gap between theoretical contributions and practical implications is a significant drawback. Finally, there are ambiguities in the definitions of the proposed metrics (e.g., the relationship between past and future reachability), and some experimental results lack sufficient explanation or justification.

The most critical reasons for the recommendation to reject this paper are the lack of experimental diversity, insufficient justification of the practical implications of the metrics, and the significant gap between theoretical assumptions and real-world applicability. While the paper demonstrates a well-developed theoretical framework, its limitations in empirical validation and practical relevance prevent it from making a strong case for acceptance.

**Additional Comments On Reviewer Discussion:**

During the rebuttal period, the authors engaged actively with the reviewers, addressing several points of criticism. Reviewer WsZQ appreciated the rigor of the framework but raised concerns about the assumption of static embeddings and the lack of experiments on diverse datasets. While the authors provided theoretical justifications for the assumptions and emphasized the domain-agnostic nature of their framework, they did not conduct additional experiments on other datasets, which remained a major concern. Reviewer FqFb raised concerns about the generalizability of the framework and the lack of a discussion on differences across recommendation models. The authors clarified that the framework is theoretically applicable across models, but the empirical evaluation did not adequately substantiate this claim. Reviewer C1ia questioned the motivation and real-world implications of the metrics, as well as specific experimental results, such as contradictions in stability metrics. The authors provided detailed responses but did not fully resolve the underlying concerns about practical relevance and experimental scope. Finally, Reviewer f6Sk highlighted ambiguities in metric definitions and concerns about oversimplifications in experimental setups. While the authors offered explanations for some of these points, critical issues such as dataset limitations and the impact of simplifying assumptions on practical auditing remained inadequately addressed.

In weighing these discussions, it became evident that while the authors made efforts to clarify their theoretical contributions, they did not sufficiently address the reviewers’ primary concerns regarding empirical validation, generalizability, and practical relevance. These unresolved issues ultimately outweighed the strengths of the paper, leading to the decision not to accept this submission.

---

### Decision · Program_Chairs · 2025-01-22

Reject